The comparative osteology of Plesiochelys bigleri n. sp., a new coastal marine turtle from the Late Jurassic of Porrentruy (Switzerland)

Püntener Christian 1 christian.puntener@jura.ch
http://orcid.org/0000-0001-5539-8691 Anquetin Jérémy 2 3
Billon-Bruyat Jean-Paul 1
1 Section d’archéologie et paléontologie, Office de la culture, République et Canton du Jura , Porrentruy , Switzerland
2 JURASSICA Museum , Porrentruy , Switzerland
3 Department of Geosciences, University of Fribourg , Fribourg , Switzerland
Young Mark
Electronic publication date: 2017 Jun 28
Publication date: 2017
Volume: 5
Electronic Location ID: e3482
Received 2017 Mar 15; Accepted 2017 May 31
Copyright: © 2017 Püntener et al.
Copyright year: 2017
Copyright holder: Püntener et al.
License: This is an open access article distributed under the terms of the Creative Commons Attribution License, which permits unrestricted use, distribution, reproduction and adaptation in any medium and for any purpose provided that it is properly attributed. For attribution, the original author(s), title, publication source (PeerJ) and either DOI or URL of the article must be cited.
License URL: https://creativecommons.org/licenses/by/4.0/

Keywords: Plesiochelys, Plesiochelyidae, Testudines, Kimmeridgian, Switzerland

Funding: Federal Roads Office (FEDRO, 95%) Republic and Canton of Jura (RCJU, 5%) The PAL A16 team is funded by the Federal Roads Office (FEDRO, 95%) and the Republic and Canton of Jura (RCJU, 5%). The funders had no role in study design, data collection and analysis, decision to publish, or preparation of the manuscript.

==============================
Background

During the Late Jurassic, several groups of eucryptodiran turtles inhabited the shallow epicontinental seas of Western Europe. Plesiochelyidae are an important part of this first radiation of crown-group turtles into coastal marine ecosystems. Fossils of Plesiochelyidae occur in many European localities, and are especially abundant in the Kimmeridgian layers of the Swiss Jura Mountains (Solothurn and Porrentruy). In the mid-19th century, the quarries of Solothurn (NW Switzerland) already provided a large amount of fossil turtles, most notably Plesiochelys etalloni, the best-known plesiochelyid species. Recent excavations in the Porrentruy area (NW Switzerland) revealed new fossils of Plesiochelys, including numerous well-preserved shells with associated cranial and postcranial material.

Methods/results

Out of 80 shells referred to Plesiochelys, 41 are assigned to a new species, Plesiochelys bigleri n. sp., including a skull–shell association. We furthermore refer 15 shells to Plesiochelys etalloni, and 24 shells to Plesiochelys sp. Anatomical comparisons show that Plesiochelys bigleri can clearly be differentiated from Plesiochelys etalloni by cranial features. The shell anatomy and the appendicular skeleton of Plesiochelys bigleri and Plesiochelys etalloni are very similar. However, a statistical analysis demonstrates that the thickness of neural bones allows to separate the two species based on incomplete material. This study furthermore illustrates the extent of intraspecific variation in the shell anatomy of Plesiochelys bigleri and Plesiochelys etalloni.

Introduction

The first radiation of crown-group turtles into marine environments occurred in Western Europe during the Late Jurassic. At that time, several groups of basal pan-cryptodiran turtles, traditionally referred to the families Plesiochelyidae Baur (1888), Thalassemydidae Zittel (1889), and Eurysternidae Dollo (1886), colonized coastal ecosystems from restricted lagoons to more open seaways. These turtles eventually disappeared at the Jurassic–Cretaceous boundary following major sea-level changes that restricted their habitats (Bardet, 1994, 1995; Bardet et al., 2014), although some phylogenetic analyses suggest that eurysternids may have survived into the Cretaceous in the form of Sandownidae or even Protostegidae (Joyce, 2007).

Plesiochelyids are relatively large coastal marine turtles known from the Late Jurassic of Switzerland, France, Germany, England, Spain, and Portugal. They are notably characterized by a fully ossified carapace and a series of derived basicranial features (Gaffney, 1975a, 1976; Anquetin, Püntener & Billon-Bruyat, 2015). Plesiochelys etalloni (Pictet & Humbert, 1857) is undoubtedly the best known plesiochelyid turtle, thanks notably to numerous specimens found in Solothurn, Switzerland (Rütimeyer, 1873; Bräm, 1965; Gaffney, 1975a, 1976; Anquetin, Püntener & Billon-Bruyat, 2014, 2015). Plesiochelys etalloni is known by both skulls and shells from the Kimmeridgian of the Swiss and French Jura Mountains, Southern England, and Northwestern Germany (Pictet & Humbert, 1857; Rütimeyer, 1873; Bräm, 1965; Karl et al., 2007; Anquetin, Deschamps & Claude, 2014; Anquetin, Püntener & Billon-Bruyat, 2014, 2015; Anquetin & Chapman, 2016). Plesiochelys planiceps (Owen, 1842), the only other valid species in the genus, is known only from a single specimen (cranium, mandibule, and remains of the hyoids and cervical vertebrae) from the Tithonian of the Isle of Portland, UK. The cranium of Plesiochelys planiceps differs in many aspects from that of Plesiochelys etalloni (Anquetin, Püntener & Billon-Bruyat, 2015).

In the present study, we describe new material of Plesiochelys from the Kimmeridgian of Porrentruy, Canton Jura, Switzerland. The new specimens were found by the Paleontology A16 project, which rescued the paleontological material discovered during the construction of the A16 Transjurane highway. These excavations yielded a great number of fossil vertebrates from Kimmeridgian layers, notably including dinosaur trackways (Marty & Hug, 2003; Marty et al., 2007; Marty, 2008; Marty & Billon-Bruyat, 2009) and numerous coastal marine turtles (Billon-Bruyat, 2005a). The rich and diverse turtle fauna from Porrentruy notably includes different species of Plesiochelyidae and Thalassemydidae (see Geological Setting) (Püntener et al., 2014; Anquetin, Püntener & Billon-Bruyat, 2015; Püntener, Anquetin & Billon-Bruyat, 2015).

As in Solothurn, Plesiochelys is by far the most common turtle taxon in the Kimmeridgian of Porrentruy. Slightly more than 100 relatively complete, but mostly disarticulated shells were discovered during the excavations, out of which 80 can be referred to Plesiochelys. Among these shells, 41 are herein assigned to a new species, Plesiochelys bigleri n. sp. We furthermore refer 15 shells to Plesiochelys etalloni, and 24 shells to Plesiochelys sp. This material is described in detail herein. The shell of Plesiochelys bigleri shows only minor anatomical differences with that of Plesiochelys etalloni, but cranial anatomy clearly distinguishes the two species. Two skulls, one associated with a shell and the other found isolated, are known for Plesiochelys bigleri. A statistical analysis confirms that the thickness of neural bones allows to separate the two species and to tentatively identify otherwise indeterminate specimens. Based on abundant shell material, the intraspecific variations in both species are discussed in details. Finally, we fully describe and illustrate elements of the appendicular skeleton, which are otherwise rarely described in the literature, hoping that this will facilitate future comparisons.

Materials and Methods

Material

The present study is based on a collection of 80 relatively complete, but mostly disarticulated shells (Table 1), most of which were found in a single stratigraphical layer in the Porrentruy region (see Geological Setting, below). Forty-one shells are referred to a new species, Plesiochelys bigleri n. sp. One of these specimens (MJSN TCH007-252) is a skull–shell association, which we designate as the holotype of the new species. An isolated cranium (MJSN TCH006-1451) is also referred to this new species and designated as its paratype.

Table 1 The new Plesiochelys material from Porrentruy.

Plesiochelys bigleri	Plesiochelys etalloni	Plesiochelys sp.	
MJSN BSY006-307 (c, pl, p)	MJSN BSY006-347 (c, pl, s, p)	MJSN BSY003-1 (c, pl, s)	
MJSN BSY006-326 (c, pl)	MJSN BSY006-376 (c, pl)	MJSN BSY008-240 (c, pl)	
MJSN BSY007-147 (c, pl, p)	MJSN BSY007-205 (c, pl)	MJSN BSY008-484 (c, pl, s, fl, p)	
MJSN BSY007-257 (c, pl)	MJSN BSY009-694 (c, pl)	MJSN BSY008-674 (c, pl)	
MJSN BSY008-206 (c, pl)	MJSN SCR003-1011 (c)	MJSN BSY009-171 (c, pl)	
MJSN BSY008-242 (c, pl)	MJSN SCR008-33 (c)	MJSN BSY009-310 (c, pl, s, p)	
MJSN BSY008-512 (c, pl)	MJSN SCR010-382 (c, pl)	MJSN BSY009-619 (c, pl)	
MJSN BSY008-567 (c, pl, s, fl)	MJSN SCR011-415 (c, pl)	MJSN SCR010-413 (pl)	
MJSN BSY008-848 (c, pl)	MJSN TCH005-216 (c, pl)	MJSN SCR010-450 (c, pl)	
MJSN BSY009-639 (c, pl)	MJSN TCH005-332 (c)	MJSN SCR010-479 (c, pl, p)	
MJSN BSY009-743 (pl)	MJSN TCH005-457 (c, pl)	MJSN SCR010-559 (c)	
MJSN BSY009-815 (c, pl, s, p)	MJSN TCH006-574 (c, pl)	MJSN SCR010-560 (c, pl)	
MJSN BSY009-892 (c, pl)	MJSN TCH007-265 (c, pl)	MJSN SCR010-561 (pl)	
MJSN CRT007-2 (c, pl)	MJSN TCH007-505 (c, pl)	MJSN SCR010-562 (c, pl)	
MJSN SCR010-327 (c, pl)	MJSN TCH007-771 (c, pl)	MJSN SCR011-111 (c, pl)	
MJSN SCR010-342 (c, pl)		MJSN SCR011-525 (c, pl, p, hl)	
MJSN SCR010-1009 (c, pl)		MJSN TCH005-286 (c, pl)	
MJSN SCR010-1047 (c, pl)		MJSN TCH005-817 (c)	
MJSN SCR010-1196 (c, pl)		MJSN TCH006-776 (c, pl, fl)	
MJSN SCR010-1279 (c, pl, hl)		MJSN TCH006-787 (c)	
MJSN SCR011-30 (c, pl, s, p, cv)		MJSN TCH007-62 (c, pl)	
MJSN SCR011-37 (c, pl)		MJSN TCH007-272 (c, pl)	
MJSN SCR011-140 (c, pl)		MJSN TCH007-541 (c, pl)	
MJSN SCR011-148 (c, pl, s, p)		MJSN TCH007-580 (pl, fl, p)	
MJSN SCR011-160 (c, pl)			
MJSN SCR011-276 (c, pl)			
MJSN SCR011-413 (c, pl, p)			
MJSN TCH005-16 (c, pl)			
MJSN TCH005-21 (c, pl, hl, cv)			
MJSN TCH005-42 (c, pl, s)			
MJSN TCH005-464 (c, pl)			
MJSN TCH005-819 (c, pl)			
MJSN TCH006-145 (c, pl)			
MJSN TCH006-767 (c, pl, p)			
MJSN TCH006-1420 (c, pl, p)			
MJSN TCH006-1451 (sk)			
MJSN TCH007-252 (sk, c, pl, s, fl, p)			
MJSN TCH007-371 (pl)			
MJSN TCH007-516 (c)			
MJSN TCH007-519 (c, pl, s, p)			
MJSN VTT006-299 (c, pl)			
MJSN VTT006-579 (c, pl)			
Notes:

Forty-two specimens are attributed to the new species Plesiochelys bigleri, 15 specimens to Plesiochelys etalloni, and 24 specimens to Plesiochelys sp. c, carapace; cv, cervical vertebra; fl, forelimb; hl, hindlimb; p, pelvis; pl, plastron; s, scapula; sk, skull. All specimens are housed at the MJSN.

Fifteen out of the 80 aforementioned shells are identified as Plesiochelys etalloni. The identification of the remaining 24 shells is uncertain because they lack diagnostic features. These specimens are therefore referred to Plesiochelys sp. However, a tentative identification of some of these specimens is provided herein based on the statistical analysis of neural bone thickness (see below).

Geological setting

All of the specimens were collected between 2001 and 2011 near the small town of Courtedoux, along the A16 Transjurane highway in the Ajoie Region, Canton of Jura, NW Switzerland (Fig. 1). The majority of the specimens come from the Lower Virgula Marls (Reuchenette Formation, Chevenez Member; Comment et al., 2015) of the sites of Bois de Sylleux (BSY; 47°24′52.22″N 7°01′21.99″E), Sur Combe Ronde (SCR; 47°24′00.3″N 7°01′35.98″E), and Tchâfouè (TCH; 47°24′14.01″N 7°01′21.57″E) (Fig. 2). These sites yielded a rich and diverse coastal marine assemblage, including invertebrates (bivalves, gastropods, cephalopods, crustaceans, and echinoderms), vertebrates (chondrichthyans, osteichthyans, turtles, crocodilians, and pterosaurs), and wood remains (Billon-Bruyat, 2005a, 2005b; Marty & Billon-Bruyat, 2009; Philippe et al., 2010; Schaefer, 2012; Comment et al., 2015; Koppka, 2015; Leuzinger et al., 2015).

Figure 1 Geographical map of the Ajoie Region, Canton Jura, Switzerland.

The five excavation sites Bois de Sylleux (BSY), Crat (CRT), Sur Combe Ronde (SCR), Tchâfouè (TCH), and Vâ Tche Tchâ (VTT) are situated along the Transjurane A16 highway (gray).

Figure 2 Stratigraphic section of the Reuchenette Formation.

Most specimens were discovered within the Lower Virgula Marls (sites of BSY, SCR, and TCH). One specimen comes from dinosaur track-bearing tidal laminites (CRT), and two from the Banné Marls (VTT). Scheme modified after Comment et al. (2015).

Plesiochelys is the dominating turtle taxon in the Lower Virgula Marls. Other taxa occur only in small numbers, including Tropidemys langii Rütimeyer, 1873, Portlandemys gracilis Anquetin, Püntener & Billon-Bruyat, 2015, Thalassemys hugii Rütimeyer, 1873, and Thalassemys bruntrutana Püntener, Anquetin & Billon-Bruyat, 2015 (Püntener et al., 2014; Anquetin, Püntener & Billon-Bruyat, 2015; Püntener, Anquetin & Billon-Bruyat, 2015). The Lower Virgula Marls are dated from the Eudoxus ammonite zone (early late Kimmeridgian; Comment et al., 2015) and are therefore slightly older than the Solothurn Turtle Limestone, which forms the uppermost member of the Reuchenette Formation and is dated from the Autissiodorensis ammonite zone (Meyer, 1994; Comment, Ayer & Becker, 2011).

One specimen of Plesiochelys bigleri (MJSN CRT007-2) has been discovered within the dinosaur track-bearing tidal laminites of the Crat site (CRT; 47°23′55.8″N 7°01′43.65″E; Fig. 2) (Billon-Bruyat et al., 2012). These laminites represent the lowermost layers of the Corbis Limestones and are dated from the Cymodoce ammonite zone (late early Kimmeridgian; Comment et al., 2015). Two other specimens of Plesiochelys bigleri (MJSN VTT006-299 and MJSN VTT006-579) are stratigraphically slightly older and come from the Banné Marls of the Vâ Tche Tchâ site (VTT; 47°25′10.13″N 7°01′04.44″E; Cymodoce ammonite zone; Fig. 2), where Tropidemys langii is the dominating turtle taxon (Püntener et al., 2014).

Most turtle shells from Porrentruy have been discovered in a state of partial disarticulation, which contrasts with the mostly articulated shells found in Solothurn. The latter were apparently rapidly incorporated in a calcareous mud, while the turtle remains from Porrentruy remained for a longer period on top of the sediment before being completely buried. This is confirmed by the common presence of incrusting bivalves (oysters) on the bone remains from Porrentruy. However, disturbance by predators or water movements was relatively limited since disarticulated elements were usually found relatively close together (Fig. 3).

Figure 3 MJSN TCH006-1420, Plesiochelys bigleri (Kimmeridgian, Porrentruy, Switzerland).

Field photograph of the specimen embedded in the Lower Virgula Marls.

Anatomical comparisons

As far as the cranium is concerned, Plesiochelys bigleri was compared to all plesiochelyids for which that part of the skeleton is known: Plesiochelys etalloni (NMB 435, NMS 8738, NMS 8739, NMS 8740, NMS 9145, NMS 40870, NMS 40871, and NHMUK R3370), Plesiochelys planiceps (OUMNH J.1582), Portlandemys mcdowelli Gaffney, 1975a (NHMUK R2914, NHMUK R3164), and Portlandemys gracilis (MJSN BSY009-708). When pertinent, comparisons were also extended to PIMUZ A/III 514, a skull–shell association from the Tithonian of the Isle of Oléron (Department of Charente-Maritime, France) initially referred to Thalassemys moseri Bräm, 1965 by Rieppel (1980), but which was recently designated as the holotype of a new taxon, Jurassichelon oleronensis Pérez-García, 2015. All of these specimens have been studied first hand by the second author. The reader is referred to the primary literature describing these specimens (Parsons & Williams, 1961; Gaffney, 1975a, 1976; Rieppel, 1980; Anquetin, Püntener & Billon-Bruyat, 2015; Anquetin & Chapman, 2016). Anatomical descriptions in the present study follow the nomenclature established by Gaffney (1972, 1979) as updated by Rabi et al. (2013).

Anatomical descriptions of shell material follow the nomenclature established by Zangerl (1969). Shell and non-shell postcranial material of Plesichelys bigleri was compared, when pertinent, to plesiochelyids (Plesichelys etalloni, Tropidemys langii, and Craspedochelys jaccardi) and thalassemydids (Thalassemys bruntrutana and Thalassemys hugii).

Statistical analysis

The length and thickness of 119 neurals 2–5 pertaining to 43 selected specimens from Porrentruy (25 Plesiochelys bigleri, 8 Plesiochelys etalloni, and 10 Plesiochelys sp.) were measured in order to test the hypothesis that neural bones are significantly thinner in Plesiochelys bigleri relative to Plesiochelys etalloni (see Table S1). Length was measured as the maximal length on the dorsal surface of the neural bone. Thickness was measured on the left and right sides and approximately at the mid-length of each neural bone (see below). The mean of these two measurements was used as the thickness value for each individual neural bone (Table S1). Measurements were taken using a digital Vernier caliper by a single operator (Christian Püntener).

Neural shape can be relatively variable within a single individual. For example, one neural can be disproportionately shorter, longer, thinner, or thicker in a given neural series. Specimens exhibiting extreme divergences from the common condition were not measured. For specimens included in this analysis, the mean neural length and mean neural thickness were computed for each individual, which had the effect of smoothing intra-individual discrepancies. The analyzed dataset therefore consists of the mean neural length, mean neural thickness, and corresponding length/thickness ratio measured for the 43 included specimens (see below).

The statistical analysis was run using PAST 3.14 (Hammer, Harper & Ryan, 2001). Length and thickness were plotted in a 2D space, whereas length, thickness, and length/thickness ratio were tested for equal medians using a non-parametric Mann–Whitney test (with Monte Carlo permutations). A discriminant analysis was also performed on the length and thickness measurements and used to tentatively identify indeterminate specimens (Plesiochelys sp.; see below).

3D models

3D models of the holotype (MJSN TCH007-252) and paratype (MJSN TCH006-1451) crania have been computed with the photogrammetry software Agisoft Photoscan 1.0.4 Standard Edition using sets of high-quality photographs of the specimens. We followed the procedures recently described by Mallison & Wings (2014). These models are provided herein as 3D PDFs (reduced resolution; to be opened with Adobe Acrobat): MJSN TCH007-252 (Fig. S1), and MJSN TCH006-1451 (Fig. S2). Scaled and textured high-resolution meshes in PLY format are also available freely on figshare (http://figshare.com/authors/J_r_my_Anquetin/651097).

A 3D surface scan of the pelvis preserved with specimen MJSN BSY006-307 was produced with an Artec Space Spider scanner (Artec Group, Luxembourg; http://www.artec3d.com) and reconstructed with Artec Studio 10, the native scanner software. The textured 3D mesh in PLY format is freely available on figshare (http://figshare.com/authors/J_r_my_Anquetin/651097).

Nomenclatural act

The electronic version of this article in portable document format will represent a published work according to the International Commission on Zoological Nomenclature (ICZN), and hence the new name contained in the electronic version is effectively published under that Code from the electronic edition alone. This published work and the nomenclatural act it contains have been registered in ZooBank, the online registration system for the ICZN. The ZooBank LSIDs (life science identifiers) can be resolved and the associated information viewed through any standard web browser by appending the LSID to the prefix http://zoobank.org/. The LSID for this publication is: urn:lsid:zoobank.org:pub:C5AE9DE8-9911-4CFD-AD09-65850C35BDEC. The online version of this work is archived and available from the following digital repositories: PeerJ, PubMed Central and CLOCKSS.

Systematic Paleontology

TESTUDINES Batsch, 1788

PAN-CRYPTODIRA Joyce, Parham & Gauthier, 2004

EUCRYPTODIRA Gaffney, 1975c

PLESIOCHELYIDAE Baur, 1888

PlesiochelysRütimeyer, 1873

Type species: Plesiochelys solodurensis Rütimeyer, 1873.

Included valid species: Plesiochelys planiceps (Owen, 1842); Plesiochelys etalloni (Pictet & Humbert, 1857); Plesiochelys bigleri n. sp.

Referred material and range: Kimmeridgian of Switzerland, France, Germany, and England (Pictet & Humbert, 1857; Maack, 1869; Rütimeyer, 1873; Oertel, 1924; Bräm, 1965; Karl et al., 2007; Anquetin, Deschamps & Claude, 2014; Anquetin, Püntener & Billon-Bruyat, 2014, 2015; Anquetin & Chapman, 2016), and Tithonian of England (Owen, 1842; Gaffney, 1975a, 1976; Anquetin, Püntener & Billon-Bruyat, 2015). Indeterminate specimens are also signaled from the Kimmeridgian–Tithonian of Portugal (Pérez-García et al., 2008) and late Tithonian of Spain (Pérez-García, Scheyer & Murelaga, 2013).

Diagnosis: See Anquetin, Püntener & Billon-Bruyat (2014, 2015).

Plesiochelys etalloni (Pictet & Humbert, 1857)

Synonymy: Emys Etalloni Pictet & Humbert, 1857 (original description); Stylemys hannoverana Maack, 1869 (subjective synonymy); Plesiochelys langii Rütimeyer, 1873 (subjective synonymy); Plesiochelys sanctaeverenae Rütimeyer, 1873 (subjective synonymy); Plesiochelys solodurensis Rütimeyer, 1873 (subjective synonymy); Plesiochelys solodurensis var. langenbergensis Oertel, 1924 (subjective synonymy).

Type material: MAJ 2005-11-1, a shell missing a large part of the carapace medially.

Illustrations of type: Pictet & Humbert (1857: plates I–III); Anquetin, Deschamps & Claude (2014: Figs. 1 and 2; Fig. S2); Anquetin, Püntener & Billon-Bruyat (2014: Figs. 2A–2D).

Type horizon and locality: “Forêt de Lect” (Lect is a small village) near Moirans-en-Montagne (Department of Jura, France), Late Jurassic. See Anquetin, Deschamps & Claude (2014) for details.

Referred material and range: Kimmeridgian of Oker and Hannover, Lower Saxony, Germany (Maack, 1869; Oertel, 1924; Karl et al., 2007); Kimmeridgian of Solothurn, Canton of Solothurn, and Glovelier and Porrentruy, Canton of Jura, Switzerland (Rütimeyer, 1873; Bräm, 1965; Gaffney, 1975a, 1976; Anquetin, Deschamps & Claude, 2014; Anquetin, Püntener & Billon-Bruyat, 2014, 2015; Table 1); Kimmeridgian of England, UK (Anquetin & Chapman, 2016).

Emended diagnosis: Plesiochelys etalloni differs from other Plesiochelys spp. in a more extensive flooring of the cavum acustico–jugulare by the pterygoid, the complete ossification of the pila prootica, and a narrow, slit-like foramen nervi trigemini. In addition, Plesiochelys etalloni differs from Plesiochelys planiceps in a smaller size, a lower lingual ridge on the maxilla, a narrower distance between the lingual ridges of the maxilla at the level of the pterygoid–vomer suture, a more rounded foramen palatinum posterius, a parietal–quadrate contact posterior to the foramen nervi trigemini, a less developed processus trochlearis oticum, a superficial canalis caroticus internus often remaining partly open ventrally, a reduced contribution of the exoccipital to the condylus occipitalis, and the anterior portion of the lingual ridge on the dentary curving medially, and from Plesiochelys bigleri in a higher temporal skull roof, a deeper pterygoid fossa, a more developed processus trochlearis oticum, an anterior foramen nervi abducentis opening more posteriorly relative to the base of the processus clinoideus, foramina anterius canalis carotici cerebralis opening almost vertically below the dorsum sellae and usually more closely set, a processus paroccipitalis extending mainly posteriorly, an increased neural and costal bone thickness, the presence of epiplastral bulbs, and a more rounded or pointed anterior margin of the anterior plastral lobe.

Plesiochelys bigleri sp. nov.

urn:lsid:zoobank.org:act:9A8EF46E-7DAA-4F5B-B727-58C559BA503C

Figs. 3–8, 9A, 9B, 10–15

Figure 4 MJSN TCH007-252, holotype of Plesiochelys bigleri (Kimmeridgian, Porrentruy, Switzerland).

Cranium in dorsal (A, B), ventral (C, D), anterior (E, F), posterior (G, H), right lateral (I, J), and left lateral (K, L) views. Gray areas indicate disarticulated sutural surfaces. Hatchings represent damaged areas. ap, antrum postoticum; bo, basioccipital; bs, basisphenoid; ccc, canalis caroticus cerebralis; cci, canalis caroticus internus; cs, crista supraoccipitalis; ct, cavum tympani; ds, dorsum sellae; epi, epipterygoid; ex, exoccipital; fce, fossa cartilaginis epipterygoidei; feng, foramen externum nervi glossopharyngei; fna, foramen nervi abducentis; fnh, foramen nervi hypoglossi; fnt, foramen nervi trigemini; fo, fenestra ovalis; fp, fenestra perilymphatica; fst, foramen stapedio-temporale; ica, incisura columellae auris; op, opisthotic; pa, parietal; pi, processus interfenestralis; pcl, processus clinoideus; ppe, processus pterygoideus externus; pr, prootic; pt, pterygoid; ptf, pterygoid fossa; pto, processus trochlearis oticum; qj, quadratojugal; qr, quadrate ridge; qu, quadrate; so, supraoccipital; sq, squamosal; tra, trabecula; un. f., unnamed foramen.

Figure 5 MJSN TCH006-1451, paratype of Plesiochelys bigleri (Kimmeridgian, Porrentruy, Switzerland).

Cranium in dorsal (A, B), ventral (C, D), posterior (E, F), and right lateral (G, H) views. Hatchings represent damaged areas. ap, antrum postoticum; bo, basioccipital; bs, basisphenoid; cci, canalis caroticus internus; cm, condylus mandibularis; cs, crista supraoccipitalis; ct, cavum tympani; ds, dorsum sellae; epi, epipterygoid; ex, exoccipital; faccc, foramen anterius canalis carotici cerebralis; faccp, foramen anterius canalis carotici palatinum; feng, foramen externum nervi glossopharyngei; fm, foramen magnum; fna, foramen nervi abducentis; fnh, foramen nervi hypoglossi; fnv, foramen nervi vidiani; fo, fenestra ovalis; fpp, foramen palatinum posterius; fst, foramen stapedio-temporale; ica, incisura columellae auris; ju, jugal; lar, labial ridge; lir, lingual ridge; mx, maxilla; op, opisthotic; pal, palatine; pi, processus interfenestralis; pcl, processus clinoideus; pr, prootic; pt, pterygoid; pto, processus trochlearis oticum; qj, quadratojugal; qr, quadrate ridge; qu, quadrate; so, supraoccipital; sq, squamosal; vo, vomer.

Figure 6 MJSN TCH006-1451, paratype of Plesiochelys bigleri (Kimmeridgian, Porrentruy, Switzerland).

Dorsal view of the dorsum sellae and sella turcica region. Hatchings represent damaged areas. The lattice pattern represents matrix infilling. bs, basisphenoid; ds, dorsum sellae; epi, epipterygoid; faccc, foramen anterius canalis carotici cerebralis; faccp, foramen anterius canalis carotici palatinum; fna, foramen nervi abducentis; fnv, foramen nervi vidiani; pcl, processus clinoideus; pr, prootic; pt, pterygoid; sc, sulcus cavernosus; st, sella turcica; tra, trabecula.

Figure 7 MJSN TCH007-252, holotype of Plesiochelys bigleri (Kimmeridgian, Porrentruy, Switzerland).

(A, B) carapace; (C, D) plastron. Line width indicates natural borders (thick lines), bone sutures (medium lines), and fractures (thin lines); double lines indicate scale sulci. co, costal; ent, entoplastron; epi, epiplastron; hypo, hypoplastron; n, neural; sp, supragygal; v, vertebral scale; xi, xiphiplastron; *, intermediate element (see text).

Figure 8 Shells of Plesiochelys bigleri.

Carapace (A, B) and plastron (C, D) of specimen MJSN TCH006-1420 (Kimmeridgian, Porrentruy, Switzerland); carapace (E, F) and plastron (G, H) of specimen MJSN BSY006-307 (Kimmeridgian, Porrentruy, Switzerland); carapace (I, J) and plastron (K, L) of specimen MJSN SCR011-140 (Kimmeridgian, Porrentruy, Switzerland); carapace (M, N) of specimen MJSN TCH005-42 (Kimmeridgian, Porrentruy, Switzerland). Line width indicates natural borders (thick lines), bone sutures (medium lines), and fractures (thin lines); double lines indicate scale sulci. co, costal; ent, entoplastron; epi, epiplastron; hyo, hyoplastron; hypo, hypoplastron; nu, nuchal; pl, pleural scale; sp, supragygal; v, vertebral scale; xi, xiphiplastron; *, intermediate element (see text).

Figure 9 Neural bones of Plesiochelys bigleri and Plesiochelys etalloni (Kimmeridgian, Porrentruy, Switzerland).

Neural 4 of specimen MJSN TCH006-1420 (Plesiochelys bigleri) in dorsal (A) and lateral left view (B); neural 4 of specimen MJSN BSY006-347 (Plesiochelys etalloni) in dorsal (C) and lateral left view (D).

Figure 10 MJSN TCH005-42, Plesiochelys bigleri (Kimmeridgian, Porrentruy, Switzerland).

Left scapula in lateral (A) and medial view (B).

Figure 11 MJSN TCH007-252, holotype of Plesiochelys bigleri (Kimmeridgian, Porrentruy, Switzerland).

Right humerus in dorsal (A), anterior (B), ventral (C), and posterior view (D), left humerus in dorsal (E), anterior (F), ventral (G), and posterior view (H).

Figure 12 MJSN TCH007-252, holotype of Plesiochelys bigleri (Kimmeridgian, Porrentruy, Switzerland).

Right radius in dorsal (A), medial (B), ventral (C), and lateral view (D); right ulna in dorsal (E), medial (F), ventral (G), and lateral view (H).

Figure 13 MJSN BSY006-307, Plesiochelys bigleri (Kimmeridgian, Porrentruy, Switzerland).

Pelvis in right lateral view.

Figure 14 MJSN SCR010-1279, Plesiochelys bigleri (Kimmeridgian, Porrentruy, Switzerland).

Right femur in dorsal (A), anterior (B), ventral (C), and posterior view (D); left fibula in dorsal (E), medial (F), ventral (G), and lateral view (H).

Figure 15 MJSN TCH005-21, Plesiochelys bigleri (Kimmeridgian, Porrentruy, Switzerland).

Cervical vertebra in dorsal (A), anterior (B), left lateral (C), ventral (D), posterior (E), and right lateral view (F).

Etymology: This species is dedicated to Pierre Bigler (Villars-sur-Fontenais, Canton of Jura, Switzerland) who so skillfully prepared many of the fossil turtles from the Paleontology A16 collection, including the holotype specimen.

Holotype: MJSN TCH007-252, near complete disarticulated carapace, epiplastra, entoplastron, hypoplastra, and right xiphiplastron; posterior part of skull; proximal part of right scapular process; proximal parts of both humeri; one radius; one ulna; both ilia with acetabulum (Figs. 4, 7, 11 and 12).

Type locality and horizon: Tchâfoué (TCH), Courtedoux, near Porrentruy, Canton of Jura, Switzerland. Lower Virgula Marls, Chevenez Member, Reuchenette Formation, late Kimmeridgian, Late Jurassic (Comment, Ayer & Becker, 2011; Comment et al., 2015).

Paratype: MJSN TCH006-1451, an isolated partial cranium (Figs. 5 and 6).

Referred material and range: Late early and early late Kimmeridgian of Porrentruy, Canton of Jura, Switzerland (see Table 1).

Diagnosis: Plesiochelys bigleri differs from other Plesiochelys spp. in a lower temporal skull roof, a shallower pterygoid fossa, a reduced processus trochlearis oticum, a more rounded foramen nervi trigemini, an anterior foramen nervi abducentis opening anteromedially to the base of the processus clinoideus, and foramina anterius canalis carotici cerebralis opening more anteriorly relative to the level of the dorsum sellae. In addition, Plesiochelys bigleri differs from Plesiochelys planiceps in a smaller size, a lower lingual ridge on the maxilla, a parietal–quadrate contact posterior to the foramen nervi trigemini, a less developed processus trochlearis oticum, a superficial canalis caroticus internus that may have remained partly open ventrally, and a reduced contribution of the exoccipital to the condylus occipitalis, and from Plesiochelys etalloni in a less extensive flooring of the cavum acustico–jugulare by the pterygoid, the absence of complete ossification of the pila prootica, a processus paroccipitalis extending posterolaterally, a reduced neural and costal bone thickness, absent or poorly developed epiplastral bulbs, and a more quadrangular anterior margin of the anterior plastral lobe.

Remarks: The better part of the 42 specimens referred to Plesiochelys bigleri will be amply illustrated in the forthcoming “Catalogues du patrimoine paléontologique jurassien,” which document the numerous discoveries made by the Paleontology A16 team. Hence, the present study concentrates on illustrating the most significant specimens only.

Plesiochelys sp.

Referred material: Early late Kimmeridgian of Porrentruy, Canton of Jura, Switzerland (see Table 1).

Remarks: Twenty-four shells among the 80 studied herein lack sufficient diagnostic characters to be clearly identified as either Plesiochelys bigleri or Plesiochelys etalloni. This is no surprise considering how close these two species are in terms of shell anatomy. These 24 indeterminate specimens are therefore provisionally referred to Plesiochelys sp. Ten of these indeterminate specimens are however tentatively identified herein based on a statistical analysis of neural bone thickness (see below).

Description of Plesiochelys Bigleri

Cranium

General description

The cranium of Plesiochelys bigleri is known from two specimens. The first one is the holotype specimen (MJSN TCH007-252) and consists of the posterior part of a skull (Fig. 4) associated with a relatively complete shell and some limb and girdle elements. The parts of the skull anterior to the pterygoids (including the nasal, palatal, and orbital regions) and the lateral part of the left otic chamber are missing. For this reason, the following bones are missing from that specimen: nasal, prefrontal, frontal, postorbital, premaxilla, maxilla, vomer, and palatine. Post-mortem deformation is minor, but the basicranium is partly disarticulated along the basisphenoid–pterygoid suture. The preservation of this specimen is not optimal. The skull was initially heavily encrusted by ferruginous mineralizations. Although most of these mineralizations were skillfully removed during preparation, sutures remain rather difficult to see in this specimen. The length of the skull as measured from the pterygoid–vomer suture to the stem of the condylus occipitalis (the condyle itself is missing) is 38.5 mm, whereas the width at the level of the condyli mandibularis is 60.6 mm (Table 2).

Table 2 Length and width measurements of the skull in Plesiochelys bigleri.

Specimen	Total length from condylus occipitalis to tip of the snout (mm)	Length from pt–vo/pal suture to condylus occipitalis (mm)	Width at the level of the condyli mandibularis (mm)	
TCH007-252	–	38.5a	60.6	
TCH006-1451	59.8a	34.4	60.0	
Notes:

These measurements should be compared with those of other plesiochelyid skulls (Anquetin, Püntener & Billon-Bruyat, 2015: Table 1).

a Specimen incomplete.

The second specimen (MJSN TCH006-1451) is an isolated, partial skull missing all of the skull roof, the ethmoid region, the orbital area, the anterior part of the snout, and most of the palate (Fig. 5). As a result, the following bones are missing from that specimen: nasal, prefrontal, frontal, postorbital, and premaxilla. The skull has been severely flattened dorsoventrally during fossilization. Crushing forces were not oriented exactly dorsoventrally and resulted in a slight lean toward the right hand side, mostly apparent in posterior view (Figs. 5E and 5F). As preserved, the skull is 59.8 mm in length from the anteriormost part of the maxilla to the condylus occipitalis (34.4 mm from pterygoid–vomer suture to condylus occipitalis; see Table 2). The width taken at the level of the condyli mandibularis is 60 mm. MJSN TCH006-1451 is therefore slightly smaller than the holotype specimen (Table 2). Compared to other plesiochelyids, MJSN TCH006-1451 is about the same size as NMS 8738 and NMS 9145, both referred to Plesiochelys etalloni, but it is much smaller than OUMNH J.1582, the holotype skull of Plesiochelys planiceps (Anquetin, Püntener & Billon-Bruyat, 2015: Table 1).

Parietal

The parietals are best preserved in the holotype specimen (MJSN TCH007-252), but fragments of the ventral part of the processus inferior parietalis are preserved on the right hand side of the paratype specimen (MJSN TCH006-1451). The parietals form the posterior part of the skull roof and meet one another medially (Fig. 4). The anterior and lateral contacts of the parietals are not preserved. Most of the posterior margin of each parietal is natural. The temporal skull roof is relatively low compared to other Plesiochelys spp. The upper temporal emargination largely exposes the foramen stapedio-temporale in dorsal view, but does not extend as far anteriorly as to expose the processus trochlearis oticum. This is similar to the condition in Plesiochelys etalloni (NMB 435, NMS 40870). The development of the upper temporal emargination is not well known in other plesiochelyids. The preservation of the dorsal surface of the parietals in the holotype (MJSN TCH007-252) does not allow to identify scute sulci with confidence.

As usual in plesiochelyids (Anquetin, Püntener & Billon-Bruyat, 2015), the anterior braincase wall formed mostly by the processus inferior parietalis is shorter than in most turtles as a result of the great development of the foramen interorbitale. As preserved, the ventral contacts of the parietal in Plesiochelys bigleri are as follows: epipterygoid and pterygoid anterior to the foramen nervi trigemini; pterygoid, quadrate and prootic posterior to the foramen nervi trigemini; and supraoccipital posteroventrally (Figs. 4I–4L). The parietal forms the dorsal half of the anterior margin of the foramen nervi trigemini. There, the parietal and pterygoid have a broad contact that excludes the epipterygoid from the margin of that foramen. The posterior margin of the foramen nervi trigemini is formed entirely by a posteroventral process of the parietal that reaches the quadrate and prevents a contact between the pterygoid and prootic. This configuration of the region of the foramen nervi trigemini is characteristic of plesiochelyids (except for the parietal–quadrate contact that is absent in Plesiochelys planiceps) and J. oleronensis (Anquetin, Püntener & Billon-Bruyat, 2015). In Plesiochelys bigleri, the foramen nervi trigemini is somewhat more rounded than in other plesiochelyids, in which the foramen is usually taller than wide and oval in shape. In most specimens of Plesiochelys etalloni (NMB 435, NMS 8738, NMS 40870, NMS 40871, and NHMUK R3370), the foramen nervi trigemini is very narrow and forms a slit-like opening.

Jugal

In the holotype specimen (MJSN TCH007-252), the anterior part of the right jugal is preserved as an isolated fragment. A small flake of bone (probably from the maxilla) is still attached to the ventromedial margin of the isolated jugal. In the paratype specimen (MJSN TCH006-1451), only the ventral part of the jugal is preserved, but the bone is more complete on the right hand side (Fig. 5). The jugal forms the posteroventral corner of the orbit and contacts the maxilla ventrally and the quadratojugal posteriorly. The other contacts of the jugal with surrounding elements are unclear. As in Plesiochelys etalloni, Plesiochelys planiceps, and J. oleronensis, the jugal lacks a medial process extending to meet the pterygoid and/or palatine. Similarly, the maxilla lacks a corresponding posteromedial process and the foramen palatinum posterius remains open posterolaterally. The ventral margin of the jugal and the posterior border of the maxilla show that the lower temporal emargination was relatively well developed in Plesiochelys bigleri, much like the condition in Plesiochelys etalloni, Plesiochelys planiceps and J. oleronensis (note that the condition in Portlandemys spp. is unknown).

Quadratojugal

The morphology of the quadratojugal is poorly known in Plesiochelys bigleri. Only a small part of the right quadratojugal is preserved anterior to the cavum tympani in the holotype (MJSN TCH007-252). A larger portion of the right quadratojugal is preserved in the paratype (MJSN TCH006-1451), but the bone is fragmented in several pieces and misses its dorsal and anterior margins. Anteriorly, the quadratojugal articulates with the jugal. Its ventral margin forms the posterior half of the lower temporal emargination. Posteriorly, the quadratojugal braces the external margin of the cavum tympani along an extended, curved suture with the quadrate (Figs. 4 and 5). It seems that the posteroventral process of the quadratojugal along the processus articularis of the quadrate is proportionally shorter in Plesiochelys bigleri than in Plesiochelys etalloni and Portlandemys gracilis (unknown in other plesiochelyids). Posterodorsally, the quadratojugal has a broad vertical contact with the squamosal just dorsal and slightly anterior to the level of the incisura columellae auris.

Squamosal

The squamosal is best preserved in the holotype (MJSN TCH007-252), but the paratype (MJSN TCH006-1451) provides additional information on the morphology of this element. As usual in turtles, the squamosal forms the posterolateral corner of the otic chamber, notably contributing to the formation of the antrum postoticum. The anterodorsal part of the bone is unfortunately not preserved in any specimen. The contacts of the squamosal in the temporal roof (notably with the postorbital and parietal) are therefore unknown in Plesiochelys bigleri. The other contacts of the squamosal are as follows: quadrate anteromedially and ventrally; quadratojugal anterolaterally; and opisthotic posteromedially. Compared to other turtles, the antrum postoticum of plesiochelyids is usually described as moderately developed (Gaffney, 1976; Rieppel, 1980; Anquetin, Püntener & Billon-Bruyat, 2015). In Plesiochelys bigleri, the antrum postoticum is also moderately developed, but, compared to other plesiochelyids, the cavity is deeper both medially and posterodorsally. Remarkably, the anterior margin of the antrum postoticum is formed entirely by the quadrate (MJSN TCH007-252; Fig. 4). This differs from the condition observed in Plesiochelys etalloni and Portlandemys gracilis (as well as J. oleronensis and most basal pan-cryptodires) in which the squamosal contributes to the anterolateral margin of the antrum postoticum. The posterodorsal parasagittal crest of the squamosal is well developed in Plesiochelys bigleri, notably the posterior part that forms an extended and somewhat pointed lamina (MJSN TCH007-252; Figs. 4I and 4J). A wide concavity is present on the lateral surface of the squamosal just posterior to the opening of the antrum postoticum. Posteroventrally, the squamosal contributes only to a limited extent to the rugose area for the attachment of the M. depressor mandibulae (Werneburg, 2011).

Maxilla

The maxilla is not preserved in the holotype specimen. Only the posterior part of the bone is preserved in the paratype (MJSN TCH006-1451), but about half of the original bone is present on the right hand side in that specimen (Fig. 5). As preserved, the maxilla contacts the jugal posterodorsally and the palatine posteromedially. As for the jugal, the maxilla lacks a posteromedial process that would close the foramen palatinum posterius posterolaterally. The maxilla seems to participate to the anterolateral margin of the foramen palatinum posterius both in dorsal and ventral views. The labial ridge is slender and very high, notably posteriorly. Anteriorly, the labial ridge is somewhat blunted in MJSN TCH006-1451, but this is possibly a preservational artifact. The labial ridge is separated from the lingual ridge by a deep trough. As in Plesiochelys etalloni, Plesiochelys planiceps, and Portlandemys mcdowelli (unknown in Portlandemys gracilis), the lingual ridge is broad and high. In these species, the lingual ridge is rugose, which contrasts with the condition in J. oleronensis where the summit of the lingual ridge is acute and smooth. In Plesiochelys bigleri, the lingual ridge is closer to the condition in other plesiochelyids, but the bone surface is slightly eroded. In contrast to Portlandemys mcdowelli, the lingual ridge is formed entirely by the maxilla. Due to poor preservation and the presence of markings on the bone surface, we were unable to locate the foramen supramaxillare in MJSN TCH006-1451.

Vomer

The vomer is missing in the holotype (MJSN TCH007-252) and only the posteroventral part of the bone is preserved in MJSN TCH006-1451 (only visible in ventral view). Posteriorly, the vomer reaches the pterygoids and fully separates the palatines on the ventral surface of the palate, but not on the dorsal surface (Fig. 5). This configuration is similar to that of Plesiochelys etalloni and Portlandemys mcdowelli, but differs from that of J. oleronensis. The condition in Plesiochelys planiceps is uncertain because the posteroventral part of the vomer is broken in the type and only known specimen (OUMNH J.1582).

Palatine

The palatine is only preserved in the paratype (MJSN TCH006-1451). The bone is not complete and only the posterior and lateral parts can be observed (Fig. 5). As preserved, the palatine contacts the pterygoid posteriorly, the vomer medially on the ventral surface of the palate, the other palatine medially on the dorsal surface of the palate, and the maxilla anterolaterally. Laterally, the palatine defines the anteromedial border of the foramen palatinum posterius. The outline of the foramen palatinum posterius is impossible to assess in Plesiochelys bigleri since the lateral margin of the palatine and anterolateral part of the pterygoid are incomplete in all of the specimens. The palatine was initially described as forming a small portion of the lingual ridge in Plesiochelys and Portlandemys (Gaffney, 1976). In fact, the palatine really contributes to the lingual ridge only in Portlandemys mcdowelli (condition unknown in Portlandemys gracilis). In Plesiochelys etalloni, Plesiochelys bigleri, Plesiochelys planiceps, and J. oleronensis, the palatine–maxilla suture extends along the medial base of the lingual ridge, i.e., on its dorsomedial edge, and the palatine therefore does not take part into the formation of the ridge or triturating surface.

Quadrate

On the dorsal surface of the otic chamber, the quadrate contacts the prootic anteromedially, the opisthotic posteromedially, and the squamosal posterolaterally. As usual, quadrate and prootic contribute relatively equally to the formation of the foramen stapedio-temporale, but variability exists (see right side of the holotype MJSN TCH007-252). Anterolaterally, the quadrate has a long curved suture with the quadratojugal. This suture is mostly parallel to the anterior margin of the cavum tympani, but lies anterior to it (the quadratojugal does not enter the margin of the cavum tympani). The cavum tympani of Plesiochelys bigleri is deeper than in other plesiochelyids. This is notably apparent anteroventral and posterodorsal to the incisura columellae auris. In contrast to other plesiochelyids, the lateral margin of the cavum tympani faces more posterolaterally than laterally in Plesiochelys bigleri (Figs. 4 and 5). The quadrate forms the entire anterior margin of the antrum postoticum, which is remarkable (see Squamosal). The incisura columellae auris remains open posteroventrally. The processus articularis of the quadrate is damaged in the holotype (MJSN TCH007-252), which may give the misleading impression that this process is short. This structure is better preserved in the paratype (MJSN TCH006-1451). A prominent ventrally infolding ridge occurs on the posterior surface of the processus articularis. Starting from the posterolateral corner of the condylus mandibularis, this ridge extends dorsomedially toward the incisura columellae auris increasing in height. Ventral to the incisura, the ridge thickens, and bends sharply medially. Pursuing its medial course, the ridge thins progressively, and finally merges with the posterolateral border of the pterygoid. This ventrally infolding ridge is found in all plesiochelyids, but also in J. oleronensis, Solnhofia parsonsi Gaffney, 1975b and Parachelys eichstaettensis Meyer, 1864, and is a strong argument to support the monophyly of these Late Jurassic coastal marine turtles from Europe (Anquetin, Püntener & Billon-Bruyat, 2015). The condylus mandibularis of Plesiochelys bigleri is remarkably narrow anteroposteriorly compared to other plesiochelyids. The condyle consists of two slightly concave facets separated by a wide, but shallow parasagittal furrow.

The quadrate forms about half of the moderately developed processus trochlearis oticum, which is less prominent than in Plesiochelys etalloni. On the anterior surface of the otic chamber, the quadrate contacts the prootic dorsomedially, the parietal medially, and the pterygoid ventromedially. This region is best preserved on the left hand side of the holotype MJSN TCH007-252. Inside the cavum acustico–jugulare, the quadrate forms the lateral part of the aditus canalis stapedio-temporalis and canalis stapedio-temporalis, as well as the lateral half of the posterior opening of the canalis cavernosus. Below the antrum postoticum, a rugose area occurs on the ventral surface of the skull and probably served for muscular attachment. In Plesiochelys bigleri, this area is mostly formed by the quadrate, with minor contributions from the opisthotic and squamosal.

Epipterygoid

The epipterygoid is best preserved in the holotype (MJSN TCH007-252), whereas only the ventral part of the bone is preserved in the paratype (MJSN TCH006-1451). In lateral view, the epipterygoid is a trapezoidal element located between the crista pterygoidea of the pterygoid and the processus inferior parietalis of the parietal. The epipterygoid is exposed on the medial surface of the anterior braincase wall, albeit to a more moderate extent. Posteriorly, a broad contact between the parietal and pterygoid excludes the epipterygoid from the anterior margin of the foramen nervi trigemini (Figs. 4I–4L). The same configuration seemingly occurs on the medial surface of the anterior braincase wall. Posteroventrally, a fossa cartilaginis epipterygoidei is present and prevents a contact between the epipterygoid and the quadrate. The anterolateral process of the epipterygoid that extends onto the dorsal surface of the pterygoid is well developed. A shallow furrow prolongs this process on the dorsal surface of the pterygoid in Plesiochelys bigleri (MJSN TCH006-1451). A similar process occurs in Plesiochelys etalloni and Portlandemys mcdowelli. This process is more reduced in Plesiochelys planiceps and J. oleronensis.

Pterygoid

Except for each processus pterygoideus externus, the pterygoids are complete in the paratype (MJSN TCH006-1451), but their ventral surface is somewhat abraded. Longitudinal striae extend from the pterygoid fossae posteriorly to the palatines and vomer anteriorly (Fig. 5C). In the holotype (MJSN TCH007-252), the anterior part of the pterygoids is more poorly preserved, although the left processus pterygoideus externus is complete. In contrast, the posterior part of the pterygoids (pterygoid fossa and quadrate process) is better preserved. The pterygoids are disarticulated from the basicranium in the holotype.

In ventral view, the pterygoid contacts the vomer anteromedially, the palatine anteriorly, the quadrate posterolaterally, and the basisphenoid posteromedially. Posteriorly, the pterygoid probably also contacted the basioccipital, but the sutures in this region are poorly preserved in both specimens. As preserved, the pterygoid does not seem to contact the exoccipital posteriorly, but this region is rarely well preserved in plesiochelyids and the presence/absence of this contact is probably of poor systematic value (Anquetin, Püntener & Billon-Bruyat, 2015). In ventral view, the processus pterygoideus externus is similar in development and shape to that of Plesiochelys etalloni and Plesiochelys planiceps. However, the parasagittal plate on the lateral margin of the processus pterygoideus externus is more developed in Plesiochelys etalloni than in the two other species. A distinct ridge extends posteromedially from the posterior edge of the processus pterygoideus externus to the posterolateral part of the ventrally open canalis caroticus internus. This ridge forms the medial margin of the pterygoid fossa. Compared to other plesiochelyids, J. oleronensis and eurysternids (Anquetin, Püntener & Billon-Bruyat, 2015), the pterygoid fossa of Plesiochelys bigleri stands out as being remarkably shallow. The configuration of the canalis caroticus internus is similar to the condition in Plesiochelys etalloni and J. oleronensis (see Anquetin, Püntener & Billon-Bruyat, 2015 for a review). The canalis caroticus internus is superficial and open ventrally at least along its anterior half (Figs. 4 and 5). The posterior half of the canalis caroticus internus may have been floored by a thin ventromedial flap of the pterygoid, but the preservation of the paratype (MJSN TCH006-1451) prevents a definitive conclusion. Since no flooring is preserved in any specimen, the position of the foramen posterius canalis carotici interni cannot be determined with precision. The anterior part of the canalis caroticus internus follows the basisphenoid–pterygoid suture. By comparison with Plesiochelys etalloni, the split between the palatine and cerebral branches of the internal carotid artery was probably not floored by bone, but the preservation of the specimens prevents a clear observation of the foramen posterius canalis carotici palatinum (possibly visible on the left-hand side of MJSN TCH006-1451) and foramen posterius canalis carotici cerebralis. The flooring of the cavum acustico–jugulare by the posterolateral part of the pterygoid is not very extensive. The processus interfenestralis of the opisthotic therefore remains largely visible in ventral view. For that matter, Plesiochelys bigleri is closer to the condition observed in Plesiochelys planiceps. Although this contact is now disarticulated in the holotype, there was a contact between the processus interfenestralis of the opisthotic and the pterygoid. Based on the disarticulated surface, this contact was probably not sutural. Lateral to this contact, the pterygoid forms the medial part of the floor of the posterior opening of the canalis cavernosus.

In the ethmoid region, the pterygoid forms the ventral margin of the foramen nervi trigemini, the rest of the foramen margin being formed by the parietal (Figs. 4I–4L). The pterygoid–parietal contact anterior to the foramen nervi trigemini excludes the epipterygoid from the margin of this foramen. Medial to the crista pterygoidea, the pterygoid forms the floor of the sulcus cavernosus. This region is best preserved in the paratype specimen (MJSN TCH006-1451). The foramen anterius canalis carotici palatinum opens in the anterior part of the sulcus cavernosus about halfway between the level of the foramen anterius canalis carotici interni and the tip of the trabecula. After exiting the foramen anterius canalis carotici palatinum, the palatine branch of the internal carotid artery continues forward in a groove within the floor of the sulcus cavernosus and is not obstructed anteriorly by a crest extending from the crista pterygoidea to the midline shelf of the pterygoid, as seen in some plesiochelyids (Anquetin, Püntener & Billon-Bruyat, 2015). Anterolaterally, a small foramen occurs on the dorsal surface of the pterygoid medial to the anterolateral process of the epipterygoid. This foramen may correspond to the foramen nervi vidiani, but this should be further investigated (see also Anquetin, Püntener & Billon-Bruyat, 2015). Anterior to this small foramen, the dorsal surface of the pterygoid forms a shallow, rounded depression, which likely served for the attachment of one of the eye muscles (Gaffney, 1976). Finally, as noted above (see Epipterygoid), a shallow furrow prolongs the anterolateral process of the epipterygoid on the dorsal surface of the pterygoid.

Supraoccipital

The supraoccipital contacts the parietal anteriorly, the prootic anterolaterally, the opisthotic posterolaterally, and the exoccipital posteriorly. A broad contact between the prootic and opisthotic separates the supraoccipital from the quadrate on the floor of the fossa temporalis superior (Figs. 4 and 5). This contrasts with the condition observed in Portlandemys mcdowelli and most specimens referred to Plesiochelys etalloni. Most of the crista supraoccipitalis is preserved in the holotype MJSN TCH007-252. This structure is relatively low, especially compared to Plesiochelys planiceps and Portlandemys mcdowelli. The condition in Plesiochelys etalloni is more difficult to appreciate based on the available material, but it seems to be intermediate in development between Plesiochelys bigleri and the two aforementioned species. The posterior part of the crista supraoccipitalis is broken in MJSN TCH007-252, but it is unlikely that this structure projected far behind the level of the condylus occipitalis. As usual, the supraoccipital forms the dorsal part of the foramen magnum.

Exoccipital

The exoccipital contacts the supraoccipital dorsomedially, the opisthotic laterodorsally, the basioccipital ventrally, and the processus interfenestralis of the opisthotic anterolaterally. As preserved, there is no contact between the exoccipital and pterygoid, which may be a difference with Plesiochelys etalloni, but this contact is often difficult to observe in this taxon (Anquetin, Püntener & Billon-Bruyat, 2015). An exoccipital–pterygoid contact is otherwise present in Portlandemys ssp. and J. oleronensis. In contrast to Plesiochelys etalloni, Plesiochelys planiceps, Portlandemys mcdowelli (condition unknown in Portlandemys gracilis), and J. oleronensis, the exoccipitals apparently do not meet in the floor of the foramen magnum, but this feature can present some intraspecific variability (Anquetin & Chapman, 2016). The contribution of the exoccipital to the condylus occipitalis is uncertain, but probably reduced. There are two foramina nervi hypoglossi on each side and they are formed exclusively by the exoccipital.

Basioccipital

The ventral aspect of the basioccipital is poorly preserved in all specimens. However, it is apparent that the basioccipital contacts the basisphenoid anteriorly and the pterygoid anterolaterally in this area. The tubercula basioccipitale are only moderately developed and correspond to what is known in similarly sized individuals of Plesiochelys etalloni, but not in larger specimens (Anquetin & Chapman, 2016). Posterodorsally, there is an extensive contact with the exoccipital, but the basioccipital nevertheless enters the ventral margin of the foramen magnum. It appears that the basioccipital forms most of the condylus occipitalis, which is only preserved (poorly) in MJSN TCH006-1451. In dorsal aspect, the basioccipital offers little remarkable features. The basis tuberculi basalis is relatively low, and an oval depressed area occurs in front of it. Finally, there is also a contact with the ventromedial margin of the processus interfenestralis of the opisthotic in the floor of the recessus scalae tympani.

Prootic

On the dorsal surface of the otic chamber, the prootic contacts the parietal anteromedially, the quadrate laterally and anteroventrally, the opisthotic posteriorly, and the supraoccipital posteromedially. The prootic forms the medial half of the processus trochlearis oticum, which is reduced compared to that of Plesiochelys etalloni. Anteromedially, the prootic is excluded from entering the posterior margin of the foramen nervi trigemini by an elongate descending process of the parietal (Figs. 4I–4L). A broad contact between this descending process of the parietal and the quadrate prevents a prootic–pterygoid contact in this area. This configuration is found in all plesiochelyids but Plesiochelys planiceps, as well as in J. oleronensis and possibly also in eurysternids (Anquetin, Püntener & Billon-Bruyat, 2015).

Inside the cavum acustico–jugulare, the prootic forms the anterior half of the fenestra ovalis, the medial part of the aditus canalis stapedio-temporalis, and most of the roof of the posterior opening of the canalis cavernosus. As in other plesiochelyids, the aditus canalis stapedio-temporalis is located more posteriorly than the posterior opening of the canalis cavernosus rather than in the roof of the latter as in many turtles. In this area, the prootic contacts the opisthotic posterodorsally, the pterygoid ventrally, and the quadrate laterally. A C-shaped furrow occurs on the surface of the prootic dorsolateral to the fenestra ovalis. Inside the cavum cranii, the prootic contacts the supraoccipital posterodorsally, the parietal anterodorsally, the basisphenoid ventromedially, and the pterygoid anteroventrally. In contrast to Plesiochelys etalloni, the pila prootica is not ossified. Medial to the foramen nervi trigemini, the prootic forms a well-defined recess that accommodated the ganglion of the trigeminal (V) nerve.

Opisthotic

On the dorsal surface of the otic chamber, the opisthotic contacts the prootic anteriorly, the quadrate anterolaterally, the squamosal posterolaterally, the supraoccipital medially, and the exoccipital posteromedially. The anterior contact with the prootic is relatively wide and clearly separates the supraoccipital and quadrate (Figs. 4 and 5). This condition is found also in Plesiochelys planiceps, Portlandemys gracilis, and J. oleronensis. In contrast, the prootic–opisthotic contact is reduced or absent in Portlandemys mcdowelli and most specimens of Plesiochelys etalloni. The posterior aspect of the opisthotic is complete only on the right hand side of the paratype (MJSN TCH006-1451). The extremity of the processus paroccipitalis forms a distinct crest for muscular attachment. In Plesiochelys bigleri, as well as in Portlandemys mcdowelli and J. oleronensis, the processus paroccipitalis extends posterolaterally. This contrasts with Plesiochelys etalloni, Plesiochelys planiceps, and Portlandemys gracilis in which the processus paroccipitalis extends more posteriorly. This difference changes the occipital outline as seen in dorsal view from a broad arch in Plesiochelys bigleri to a more narrow arch in Plesiochelys etalloni.

Ventrally, the opisthotic forms a large part of the roof of the cavum acustico–jugulare. In this area, the opisthotic contacts the prootic anteriorly, the quadrate anterolaterally, the squamosal posterolaterally, and the exoccipital posteromedially. The extremity of processus interfenestralis is triangular in shape and contacts the exoccipital, basioccipital, and pterygoid. The medial margin of the processus interfenestralis is pierced by a large fenestra perilymphatica, which is apparently entirely contained in bone thanks to a ventromedial contribution from the basioccipital (MJSN TCH007-252). A well-defined foramen externum nervi glossopharyngei opens at the base of the processus interfenestralis. It is noteworthy that this foramen opens more laterally than in other plesiochelyids and J. oleronensis. Lateral to the base of the processus interfenestralis and foramen externum nervi glossopharyngei, the opisthotic forms a strong curving ridge oriented anteroposteriorly. As far as we know, this ridge is not present in other plesiochelyids.

Basisphenoid

In ventral aspect, the basisphenoid contacts the pterygoid anteriorly and laterally, and the basioccipital posteriorly (Figs. 4 and 5). As noted above, the basisphenoid–pterygoid suture is disarticulated in the holotype, but this contact is better preserved in the paratype (MJSN TCH006-1451). The posterior contact with the basioccipital is poorly preserved in both specimens. The general outline of the basisphenoid in ventral view is triangular, as in most plesiochelyids (Anquetin, Püntener & Billon-Bruyat, 2015). The ventral surface of the basisphenoid is apparently slightly concave. The morphology of the canalis caroticus internus is similar to the condition observed in Plesiochelys etalloni and J. oleronensis (Anquetin, Püntener & Billon-Bruyat, 2015). The canalis caroticus internus is superficial and runs along the basisphenoid–pterygoid suture as a ventrally open canal. The posterior part of the canalis caroticus internus may have been partly floored by a ventromedial flap formed by the pterygoid, but this region is not well preserved in any specimen. Based on the available material, the position of the foramen posterius canalis carotici interni cannot be determined. As in Plesiochelys etalloni, the split between the cerebral and palatine branches of the internal carotid artery was probably not floored by bone. However, the preservation prevents a clear observation of the foramen posterius canalis carotici palatinum. In the holotype, a portion of each canalis caroticus cerebralis is preserved on the anterior part of the basisphenoid. Anteriorly, these two canals penetrate deeply in the basisphenoid and exit in the sella turcica dorsally. A small unnamed foramen opens dorsomedially along the canalis caroticus cerebralis (only visible in MJSN TCH007-252; Fig. 4).

The dorsal aspect of the basisphenoid is better preserved in the paratype (MJSN TCH006-1451) and can be readily observed in that specimen thanks to the loss of the anterior part of the roof of the cavum cranii (Fig. 6). The basioccipital contacts the basioccipital posteriorly, the prootic posterolaterally, and the pterygoid anterolaterally and anteriorly. The part of the basisphenoid that floors the cavum cranii is slightly concave. A moderately raised area located sagittally on the posterior margin of the basisphenoid corresponds to the anterior part of the basis tuberculi basalis. The posterior foramen nervi abducentis opens about midway along the part of the basisphenoid that floors the cavum cranii. The anterior foramen nervi abducentis opens ventral and slightly anteromedial to the base of the processus clinoideus, relatively close to the basisphenoid–pterygoid suture. This is the condition usually found in plesiochelyids, with the exception of Plesiochelys etalloni (Anquetin, Püntener & Billon-Bruyat, 2015). The processus clinoideus is formed just medial to the basisphenoid–prootic suture. In contrast to Plesiochelys etalloni, the pila prootica is not ossified. The dorsum sellae is high and does not overhang the sella turcica, as usual in plesiochelyids. The surface below the dorsum sellae is devoid of ridge and slopes relatively gently anteriorly. In that matter, Plesiochelys bigleri is intermediate between Plesiochelys etalloni, in which the surface below the dorsum sellae is near vertical, and Portlandemys spp., in which that surface slopes very gently anteriorly resulting in a relatively long distance between the dorsum sellae and the foramina anterius canalis carotici cerebralis (Anquetin, Püntener & Billon-Bruyat, 2015). The foramina anterius canalis carotici cerebralis are separated by a broad bar of bone, which is unusual in plesiocheyids, and open slightly posterior to the level of the foramina anterius canalis carotici palatinum. The trabeculae are relatively short and straight, and frame a small, well-defined sella turcica. The rostrum basisphenoidale is short, and the anterior tip of the trabeculae actually represents the anteriormost extension of the basisphenoid in dorsal view.

Carapace

General description

Elements of the carapace are preserved in 39 out of the 41 shells herein referred to Plesiochelys bigleri (Figs. 7 and 8). The largest specimens reach a carapace length of about 550 mm (MJSN BSY006-307, MJSN SCR011-30, and MJSN TCH005-21), whereas the smallest have a carapace length of about 220–250 mm (MJSN BSY008-848, MJSN SCR010-327, and MJSN TCH007-516). Measuring carapace length with precision in disarticulated and incomplete specimens is not possible, but we estimate that most specimens are about 450 mm in carapace length, which corresponds approximately to what is known for Plesiochelys etalloni in Solothurn. Most shells of Plesiochelys bigleri are disarticulated, but two shells (MJSN BSY006-307 and MJSN TCH007-519) were found articulated. Bones were prepared out of the marly matrix individually and we were able to reassemble them on a moldable sand bed and reconstruct the 3D shape of the shell. Post-mortem deformation occurred in most specimens and to variable extents. Our reconstructions are therefore tentative. The resulting carapace outlines are either evenly oval (e.g., MJSN TCH006-767), or roundish (e.g., MJSN TCH007-252; Figs. 7A and 7B). In some specimens the posterior carapace part is pointed (e.g., MJSN BSY009-815 and MJSN SCR011-30).

The shells of Plesiochelys bigleri and Plesiochelys etalloni are very similar. The following description therefore primarily focuses on the few differences between the two species and the intraspecific variations observed in Plesiochelys bigleri. A general description of the shell morphology of Plesiochelys etalloni and a discussion of its intraspecific variability can be found elsewhere (Anquetin, Deschamps & Claude, 2014; Anquetin, Püntener & Billon-Bruyat, 2014).

Nuchal

The shape of the nuchal in Plesiochelys bigleri varies from almost rectangular (e.g., MJSN SCR011-140; Figs. 8I and 8J) to trapezoidal (e.g., MJSN BSY007-257). This element can be about as wide as long (e.g., MJSN BSY009-815) or clearly wider than long (e.g., MJSN SCR011-160). The nuchal notch is usually shallow, but can also be much reduced (e.g., MJSN SCR011-160) or strongly pronounced (e.g., MJSN BSY007-257). In some specimens (e.g., MJSN TCH006-1420; Figs. 8A and 8B), the posteromedial part of the nuchal that rticulates with the first neural projects posteriorly.

Neurals, suprapygals, and pygal

Like Plesiochelys etalloni, Plesiochelys bigleri usually has eight neurals, one intermediate element, two suprapygals, and one pygal. However, this condition may vary. Neural bones can be split in two (e.g., neural 2 in MJSN SCR010-342), fused together (e.g., neural 8 with the intermediate element in MJSN TCH005-21), or be reduced (e.g., neural 7 in MJSN SCR011-140) or completely obliterated (e.g., neural 7 in MJSN SCR011-413 and MJSN TCH006-1420; Figs. 8A and 8B) by costals with midline contact. The first neural is rectangular to oval in shape. It usually tapers posteriorly (not in MJSN BSY007-257 and MJSN TCH005-42; Figs. 8M and 8N). Neurals 2–6 are elongated hexagons with shorter sides facing anteriorly. However, these hexagons are often deformed (e.g., MJSN BSY007-257, MJSN SCR010-342, MJSN SCR011-30, MJSN SCR011-276, MJSN SCR011-413, MJSN TCH005-42, and MJSN TCH006-767). Neurals 7 and 8 are shorter and more irregular in shape than the preceding neurals. Their growth is often constrained by a midline contact of the costals. Neural 8 is usually the smallest bone in the series. In some specimens, it is pentagonal with shorter sides facing posteriorly (e.g., MJSN BSY009-815, MJSN TCH007-252; Figs. 7A and 7B). The thickness of the neural bones in Plesiochelys bigleri is significantly reduced compared to Plesiochelys etalloni (Fig. 9). This is especially obvious for neurals 2–5. We statistically demonstrate this difference herein (see below).

As relatively common in turtles (Zangerl, 1969), the posteromedial region of the carapace is characterized by a great deal of intraspecific variation in Plesiochelys bigleri. The element that follows neural 8 corresponds to the intermediate element of Anquetin, Püntener & Billon-Bruyat (2014). This intermediate element varies much in shape, but usually tapers anteriorly (e.g., MJSN SCR011-140, MJSN SCR011-160, and MJSN SCR011-276). In one specimen (MJSN TCH006-1420; Figs. 8A and 8B) the intermediate element is fused to the first suprapygal.

Plesiochelys bigleri usually has two suprapygals that are clearly wider than long. Although irregular in shape (usually from trapezoidal to sub-triangular), the first suprapygal usually tapers anteriorly and the second posteriorly (e.g., MJSN BSY009-815, MJSN SCR011-160, MJSN TCH006-1420; Figs. 8A and 8B). The two suprapygals can fuse into a single element (e.g., MJSN TCH007-519 and MJSN SCR011-148). In some specimens, the first suprapygal is divided in two (e.g., MJSN SCR011-30, MJSN TCH007-252; Figs. 7A and 7B). The pygal is a trapezoidal element that is wider than long. Its relative size varies substantially from one individual to another.

Costals and peripherals

Plesiochelys bigleri has eight pairs of costals and eleven pairs of peripherals. Their shape and arrangement are relatively stable within the species. The length–width ratio of costal four corresponds to that of Plesiochelys etalloni and differs notably from that of C. jaccardi (Anquetin, Püntener & Billon-Bruyat, 2014). The costals in Plesiochelys bigleri are not as thick as the same elements in Plesiochelys etalloni. In the latter, the proximal part of the costals is very thick, matching the thick neural bones, and the costals usually remain relatively thick on their distal margin. In Plesiochelys bigleri, both the proximal and distal parts of the costals are thinner. A midline contact occurs frequently between costals 7 (e.g., MJSN SCR011-140; Figs. 8I and 8J), sometimes also between costals 6 (e.g., MJSN SCR011-30, MJSN TCH006-1420; Figs. 8A and 8B), and rarely between costals 8 (e.g., MJSN TCH005-42; Figs. 8M and 8N). Such a reduction of posterior neurals and midline contacts of costals can also occur in Plesiochelys etalloni (e.g., MAJ 2005-11-1 and MJSN TCH006-574), but more rarely than in Plesiochelys bigleri. This condition appears to occur more commonly in C. jaccardi (Anquetin, Püntener & Billon-Bruyat, 2014).

Scutes of the carapace

There are three cervical scutes of about equal size in Plesiochelys bigleri (best preserved in MJSN CRT007-2 and MJSN TCH007-519). However, as common in plesiochelyids (see Anquetin, Püntener & Billon-Bruyat, 2014), cervical scutes are difficult to discern in many specimens.

There are usually five vertebrals, four pairs of pleurals, and 12 pairs of marginals. Vertebrals 1 and 5 are always the shortest in the series. The second and third vertebrals are rectangular (e.g., MJSN BSY009-815) to hexagonal (e.g., MJSN SCR011-30) in shape and cover one third (e.g., MJSN TCH007-519) to about half (e.g., MJSN TCH006-767) of the costal length laterally. Vertebral 4 is usually hexagonal in shape. In some specimens it almost extends as far as the peripherals laterally, significantly reducing the size of pleural 4 (e.g., MJSN BSY006-326 and MJSN SCR011-160). The intervertebral scute sulci usually run on neurals 1, 3, and 5. The scute sulcus between vertebrals 4 and 5 most often runs on the intermediate element (e.g., MJSN BSY009-815), but can also cross neural 8 (e.g., MJSN SCR011-148) or the first suprapygal (e.g., MJSN TCH007-252; Fig. 7B). The 12th pair of marginals is either restricted to the pygal and peripheral 11 (e.g., MJSN BSY006-326 and MJSN TCH006-767), or extends on the suprapygal 2 (e.g., MJSN TCH006-1420, MJSN SCR011-140; Figs. 8B and 8J).

One specimen (MJSN TCH005-42; Figs. 8M and 8N) shows an anomalous scute pattern. It has eight partially wedged vertebrals and five irregularly sized pleurals (maybe even six on the right side). This scute pattern shows no symmetry and we consider it as an abnormal condition in Plesiochelys bigleri.

Plastron

General description

Elements of the plastron are preserved in 40 out of the 41 shells herein referred to Plesiochelys bigleri. With 481 mm, MJSN SCR10-1279 has the longest preserved plastron. In contrast, the well-preserved juvenile specimen MJSN SCR010-327 has a plastron length of only 194 mm. The anterior plastral lobe of Plesiochelys bigleri is most often somewhat quadrangular in outline (e.g., MJSN SCR010-1196, MJSN SCR011-140; Figs. 8K and 8L), but can exceptionally also be rounded (e.g., MJSN CRT007-2), or even pointed (e.g., MJSN TCH006-1420; Figs. 8C and 8D). In Plesiochelys etalloni, the anterior plastral lobe usually has a rather rounded, sometimes pointed, anterior outline (Anquetin, Püntener & Billon-Bruyat, 2014). Like in Plesiochelys etalloni, a central plastral fontanelle (always longer than wide) is occasionally present (e.g., MJSN TCH006-1420; MJSN BSY006-307; Figs. 8C, 8D, 8G and 8H).

Epiplastra and entoplastron

The shape of the epiplastra varies from rather rectangular (MJSN SCR010-1196) to triangular (e.g., MJSN TCH006-1420; Figs. 8C and 8D). The anterolateral border of the epiplastra is often somewhat angular, mirroring the anterior outline of the anterior plastral lobe. Epiplastral bulbs, as described in Plesiochelys etalloni (Bräm, 1965; Anquetin, Püntener & Billon-Bruyat, 2014), are absent or only weakly expressed (e.g., MJSN TCH007-252; Fig. 7C). The entoplastron is usually roundish (e.g., MJSN SCR011-140; Figs. 8K and 8L) to roughly kite-shaped (e.g., MJSN TCH007-252; Figs. 7C and 7D), but its relative size and length–width proportion varies greatly from one individual to another. The entoplastron of the juvenile specimen MJSN SCR010-327 is elongated and kite-shaped.

Hyo-, hypo-, and xiphiplastra

The hyoplastra are always longer than wide, although only by a small amount in a few individuals (e.g., MJSN BSY009-815). This corresponds to what is known in Plesiochelys etalloni and many other turtles, and contrasts with the condition in C. jaccardi, where the hyoplastra are clearly wider than long (Anquetin, Püntener & Billon-Bruyat, 2014). The suture between the hyo- and hypoplastra is generally straight, but sometimes shows individual symmetric anteroposterior projections (e.g., MJSN CRT007-2, MJSN BSY006-307; Figs. 8G and 8H). There is a small supernumerary bone between the hyoplastra of the juvenile specimen MJSN SCR010-327, but the sutures of this roundish bone are absent on the visceral side of the plastron. The posterolateral borders of the hypoplastra often project so as to articulate with the corresponding notches of the xiphiplastra (e.g., MJSN SCR011-413). The xiphiplastra are always longer than wide and often asymmetric. In most specimens, the xiphiplastra are relatively long elements, possibly more elongated in proportion than in Plesiochelys etalloni. However, in some individuals the xiphiplastra appear to be significantly reduced in length, but still longer than wide (e.g., MJSN BSY009-815).

Scutes of the plastron

The gular and extragular scute sulci are relatively shallow. In Plesiochelys etalloni, the gular and extragular sulci are usually deeper, notably anteriorly, as a result of the presence of the epiplastral bulbs. As in Plesiochelys etalloni, the gular scute is either restricted to the epiplastra (e.g., MJSN CRT007-2), or extends to the entoplastron (e.g., MJSN TCH007-252; Fig. 7D). In those specimens where the plastral midline sulcus is discernable, it is irregularly sinuous, sometimes creating small supernumerary scutes (e.g., MJSN SCR011-30). A similar condition is also known in Plesiochelys etalloni.

Generally, the sulcus between the humeral and pectoral scutes is straight, while the sulcus between the pectoral and abdominal scutes is curving anteriorly. The sulcus between the femoral and anal scutes is usually restricted to the xiphiplastron, but extends to the hypoplastron in some specimens. In the latter case, the anal scutes may form a rectangular anterior projection on the hypoplastra (e.g., MJSN SCR011-30). A similar variability is present in Plesiochelys etalloni (Anquetin, Püntener & Billon-Bruyat, 2014).

There are usually four pairs of inframarginal scutes. As in Plesiochelys etalloni, the third one is generally the longest in the series. It covers the suture between hyo- and hypoplastron. The juvenile specimen MJSN SCR010-327 shows a variation of this pattern by having three inframarginal scutes on the left hypoplastron. It is however unclear whether this resulted in a total of five inframarginal scutes on the left side and whether this was also the case on the right side. The inframarginal scutes usually extend laterally on the peripherals. Occasionally some of them are restricted to plastral elements (e.g., MJSN CRT007-2 and MJSN TCH006-767). This is also a condition known in Plesiochelys etalloni (Anquetin, Püntener & Billon-Bruyat, 2014).

Pectoral girdle

Eight partially preserved scapulae are associated with shells of Plesiochelys bigleri (MJSN BSY008-567, MJSN BSY009-815, MJSN SCR011-30, MJSN SCR011-148, MJSN TCH007-252, MJSN TCH007-519, MJSN TCH005-42; Fig. 10). The two scapulae of MJSN BSY008-567 are attached to the visceral side of the plastron by sediment, approximately in situ and in contact with the humeri. The other six scapulae are disarticulated.

The dorsally projecting scapular process and anteromedially projecting acromion process are elliptic cylinders. The distal parts of these processes are missing in most specimens. Only the acromion process in the right scapula of MJSN BSY008-567 is probably complete. It measures 59 mm from its distal end to the notch between the scapular process and the coracoid. The two processes form a scapular angle of 102° in MJSN TCH005-42 (Fig. 10), the only scapula of Plesiochelys bigleri where this angle can be measured with confidence. None of the scapulae associated with Plesiochelys bigleri provides clear information about the nature of the glenoid fossa or about the articular surface for the coracoid. As in other plesiochelyids, as well as thalassemydids and eurysternids, the glenoid neck is well developed.

The posteromedially projecting coracoid itself is only partially preserved in MJSN TCH005-42 (proximal part) and MJSN TCH007-519 (distal part, partially covered by bones and sediment). It is unclear whether the disarticulated coracoid of MJSN TCH005-42 belongs to the left (preserved) or right (unpreserved) scapula. The short diaphysis is irregularly cylindric in cross section. It broadens proximally where it articulates with the scapular neck. The distal end of the bone is missing, but the broadening diaphysis indicates yet the beginning of the coracoid blade. The latter is partially observable in MJSN TCH007-519, where it forms a broad blade.

The scapula is also known in Plesiochelys etalloni (NMS 8584, NMS 8731, NMS 9153, and NMB 435), as well as in Thalassemys bruntrutana and Thalassemys hugii (Püntener, Anquetin & Billon-Bruyat, 2015). The scapular angle of Plesiochelys bigleri (see above) corresponds to the angle previously measured for Plesiochelys etalloni in specimens from Solothurn and contrasts with the wider angles in Thalassemys bruntrutana and Thalassemys hugii (Püntener, Anquetin & Billon-Bruyat, 2015: Table 1). The scapula of Plesiochelys etalloni does not show any significant anatomical differences to the scapula of Plesiochelys bigleri.

Humerus

Four humeri are associated with shells of Plesiochelys bigleri (the proximal parts of both humeri of MJSN BSY008-567 and MJSN TCH007-252, respectively). While both humeri of MJSN TCH007-252 (Fig. 11) are disarticulated, the humeri of MJSN BSY008-567 are attached to the visceral side of the plastron by sediment, approximately in situ and in contact with the scapulae.

The proximal articulation is a hemispherical head that projects dorsally with approximately 135° from the horizontal plane of the humerus (only measurable in the humeri of MJSN TCH007-252). The anteriorly expanding lateral process is the smaller one, as in most cryptodires (Gaffney, 1990). A strong deltopectoral crest projects ventrally from its lateral border (best visible in the left humerus of MJSN TCH007-252; Figs. 11F–11G). The larger, posteriorly expanding medial process is only slightly bulged ventrally, so that the intertubercular fossa is mainly defined by the deltopectoral crest. More distally, the narrowing diaphysis forms a waist almost circular in section.

The humerus is also partly known in Plesiochelys etalloni (NMS 8584 and NMB 435) and Tropidemys langii (MJSN VTT006-253 and MJSN VTT010-17). The observed features of the humerus of Plesiochelys bigleri correspond in all aspects to the humerus of Plesiochelys etalloni. According to Püntener et al. (2014), the humerus of Tropidemys langii (MJSN VTT006-253) has a more expanded medial process and a broader and flatter diaphysis. However, a more recent discovery of a humerus of Tropidemys langii (MJSN VTT010-17) contradicts this statement and suggests that the observed features of MJSN VTT006-253 are probably due to postmortem flattening. On the base of the new material, the humerus of Tropidemys langii does not show significant anatomical differences to the humerus of Plesiochelys bigleri and Plesiochelys etalloni.

Radius and ulna

The radius and ulna are only known in the holotype specimen (MJSN TCH007-252; Figs. 12A–12H). The ulna can be confidently identified as a right ulna, but the radius is more poorly preserved and is only tentatively identified as a right one. The radius is a slim bone with a cylindric diaphysis of only 5 mm diameter at its narrowest point (Figs. 12A–12D). The bone is twisted, so that the proximal and distal heads stand perpendicular to each other. Although not completely preserved, it is clear that the distal expansion was broader and flatter than the proximal one, which is a common condition in turtles.

The ulna is strongly bent along the long axis due to postmortem deformation (Figs. 12F and 12H). It is clearly broader than the radius. The proximal head forms a moderately concave sigmoid notch, and the olecranon is poorly developed. Below the proximal head, the medial surface of the bone bears a well-developed bicipital tubercle (Figs. 12E–12G). The diaphysis is flat and 7 mm broad at its narrowest point. Toward the distal end, the bone remains flat and broadens up to 16 mm. The articulation for the carpals is not preserved.

Bräm (1965) briefly described one radius of Plesiochelys etalloni (NMS 8731). Unfortunately, only a fragment of this radius has been preserved into the present day, leaving only Bräm’s description as a reference. According to Bräm (1965), the distal expansion of the radius of Plesiochelys etalloni is larger and more stoutly built than the proximal expansion.

Both ulnae of the same specimen (NMS 8731) are preserved. The distal expansion of these bones is quadrangular in shape. In contrast, this part is triangular-shaped in Plesiochelys bigleri. However, it is not clear whether this is a real anatomical difference or whether it is due to an insufficient preservation of MJSN TCH007-252.

Pelvic girdle

Ten partially preserved pelves are associated with shells of Plesiochelys bigleri (MJSN BSY006-307, MJSN BSY-147, MJSN BSY009-815, MJSN SCR010-30, MJSN SCR011-148, MJSN SCR011-413, MJSN TCH006-767, MJSN TCH006-1420, MJSN TCH007-252 and MJSN TCH007-519). The following description is mainly based on the sub-complete pelvis of MJSN BSY006-307 that is still articulated in its original 3D shape, though it suffered minor postmortem deformation (Fig. 13). Right and left halves of the pelvis are still interconnected posteriorly by the ischium plate. The anterior midline connection of the pubes is not preserved. Although both acetabula are completely preserved on both sides of the pelvis, the ilium and pubis are much better preserved on the right side.

The acetabulum is a relatively deep and somewhat kidney-shaped cavity (Fig. 13). Its longer axis (about 35 mm long in MJSN BSY006-307) lies in the horizontal plane, its shorter axis (about 20 mm) in the vertical plane. The three bones of the pelvis (ilium, pubis, and ischium) form the acetabulum with their proximal parts. The sutures between these bones are best visible on the medial surface of the right acetabular region in MJSN BSY006-307.

The ilium extends posterodorsally (Fig. 13). Proximally, where it contributes to the acetabulum, its maximal width reaches 29 mm. More distally, it narrows to form an irregularly oval, slightly twisted shaft (about 15 mm wide in MJSN BSY006-307). At the distal end, the ilium is flatter and expands in the anteroposterior direction. However, the posterodorsal iliac process and the anterodorsal articulation surface with the sacral rib are only very partially preserved in Plesiochelys bigleri (MJSN TCH007-519 and MJSN SCR011-30, respectively).

The proximal part of the pubis extends anteroventrally approximately in the same axis as the ilium (Fig. 13). It narrows below the acetabulum, but the shaft remains much broader and flatter than the ilium shaft. Distally, the pubis divides into two parts: a ventrally orientated lateral pubic process and an anteriorly orientated part that normally forms the thyroid fenestra together with the ischium. The lateral pubic process is about 24 mm long in MJSN BSY006-307. At the distal end, it has an oval articulation surface (only partially preserved in MJSN BSY006-307) that rested on the dorsal surface of the xiphiplastra. The anteromedial part of the pubis is broken in MJSN BSY006-307, and the shape of the thyroid fenestra remains unclear in Plesiochelys bigleri.

The ischium extends posteroventrally, first narrowing into a short, circular shaft, then broadening again medially in order to form a broad plate that meets the other ischium medially. This ischial plate is concave dorsally. The contact between ischium and pubis at the anterior margin of this plate is not preserved in any specimen. The lateral ischial process is strong and extends posteriorly from the ischium. Due to postmortem deformation, the right process is strongly bent dorsally in MJSN BSY006-307. The posterior margin of the ischium, between the lateral ischial process and the midline contact with the other ischium, forms a shallow depression. The ventral surface of the ischial plate is generally convex. It bears a shallow V-shaped rugose area that points anteriorly.

The exceptional preservation of the pelvis of MJSN BSY006-307 is unique among plesiochelyids. Bräm (1965) described the pelvis of Plesiochelys etalloni (mainly based on NMS 8731) and C. jaccardi (based on NMS 8713–8718), but this material is strongly fragmented and deformed. The observable features correspond fairly well with the pelvis of Plesiochelys bigleri, namely the general shape of the ilium (well visible in NMS 8731). Bräm (1965) concluded that there are no significant differences between the pelvis of Plesiochelys etalloni and C. jaccardi.

Femur

Three femora are associated with shells of Plesiochelys bigleri (both of MJSN SCR010-1279 and one of MJSN TCH005-21). The femur of MJSN TCH005-21 is strongly deformed and its proximal part is badly damaged. Only the proximal head of the left femur of MJSN SCR010-1279 is preserved. The right femur of MJSN SCR010-1279 is almost completely preserved, but most of the bone surface is covered by encrustation. The following description is therefore mainly based on the right femur of MJSN SCR010-1279 (Figs. 14A–14D).

The femur of Plesiochelys bigleri is essentially a straight bone that is only slightly arched dorsally. The dorsally projecting proximal head and the ventrally projecting distal articulations give the left femur an elongated S-shape in posterior view (Fig. 14D). The right femur of MJSN SCR010-1279 is 134 mm long. The proximal head projects from the long axis of the femur at an angle of about 50° (Figs. 14B and 14D). It is hemispherical, but elongated along the long axis of the femur (about twice as long as wide), which is apparently more consistent with swimming than walking (Zug, 1971). This contrasts with the more roundish femoral head of Tropidemys langii (Püntener et al., 2014). In ventral view, the two trochanters form a deep, narrow, and V-shaped intertrochanteric fossa (Fig. 14C). The posteriorly situated trochanter major is slightly shorter than the trochanter minor (anteriorly), but expands more prominently on the horizontal plane. Lateral ridges on both trochanters form a shallow V-shaped depression just distal to the intertrochanteric fossa, giving the latter a terraced appearance.

The diaphysis is oval to circular in cross section. The bone is gradually broadening toward the distal end. The condyles are only slightly less expanded than the trochanters. The medial condyle is strongly arched and tappers proximally. The lateral condyle is not completely preserved, but it seems smaller and more roundish than the medial condyle. A deep fossa (about 6 mm deep) separates the two articulation surfaces from each other (Fig. 14C).

Within plesiochelyids, the femur is partly known in Plesiochelys etalloni (NMS 8584 and MNS 8731), C. jaccardi (NMS 8713–8718), and Tropidemys langii (MJSN VTT010-13). Based on this incomplete material, the femur of Plesiochelys bigleri, Plesiochelys etalloni, and C. jaccardi cannot be differentiated. On the other hand, these species differ from Tropidemys langii, in which the trochanters expand more prominently along the horizontal plane and the intertrochanteric fossa is shallower, wider, and more rounded at its base (Püntener et al., 2014).

Fibula

Two left fibulae are associated with shells of Plesiochelys bigleri (MJSN TCH005-21 and MJSN SCR010-1279). The proximal third of the fibula of MJSN TCH005-21 is missing. The fibula of MJSN SCR010-1279 is complete (Figs. 14E–14H). It is 84 mm long, which corresponds to 63% of the femur length. Proximally, the fibula is only slightly expanded and has a small, hemispherical articulation surface facing moderately ventrally. A swelling on the medial edge probably marks the attachement site of the proximal tibiofibular ligament (Figs. 14E–14G). The shaft is a rather flat, elliptic cylinder. At its narrowest point it has only one third of the width of the proximal expansion. Distally, the fibula is almost twice as expanded as proximally. Here the bone is slightly concave dorsally and has a somewhat triangular articulation surface facing moderately ventrally (Figs. 14E and 14G). The attachement site for the distal tibiofibular ligament is again marked by a swelling on the medial edge of the bone (Figs. 14E–14G).

The fibula is also partly known in Plesiochelys etalloni (NMS 8584 and MNS 8731) and C. jaccardi (NMS 8713–8718). Based on this incomplete and deformed material, the fibula of Plesiochelys bigleri, Plesiochelys etalloni, and C. jaccardi cannot be differentiated.

Vertebral column

Two disarticulated cervical vertebrae are associated with shells of Plesiochelys bigleri. The cervical vertebra of MJSN TCH005-21 is almost complete, but somewhat deformed (Fig. 15). The cervical vertebrae associated with MJSN SCR011-30 is more damaged and lacks three zygapophyses. The precise position of these two vertebrae in the cervical series is unclear. The centrum is moderately elongated and oval in cross section (slightly flattened dorsoventrally). There is a robust, but low ventral keel running all of the length of the centrum in MJSN TCH005-21 (Fig. 15D; not preserved in MJSN SCR011-30). The two known cervical vertebra are amphicoelous, with oval and slightly concave central articulations (Figs. 15B and 15E). The neural arch is moderately high (especially posteriorly) and the neural spine is reduced to a low longitudinal ridge (MJSN SCR011-30). The prezygapophyses and postzygapophyses are widely separated (Figs. 15A and 15D). The articular surface of the prezygapophyses is oriented dorsally and slightly medially. The anterior margin of the neural arch forms a deep embayment between the prezygapophyses (Fig. 15A). The articular surface of the postzygapophyses faces ventrally and slightly laterally. A strong V-shaped ridge occurs on the dorsal surface of the postzygapophyses and defines a deep triangular fossa between the postzygapophyses (Figs. 15A and 15E). The transverse processes is poorly developed and is situated anteriorly along the centrum (MJSN SCR011-30).

Cervical vertebrae are known in Plesiochelys etalloni (NMS 8584), C. jaccardi (NMS 8713–8718), and Thalassemys hugii (NMS 8595–8609). The cervical vertebrae of Plesiochelys etalloni and C. jaccardi are strongly deformed and broken. The few discernable features are consistent with what is known in Plesiochelys bigleri. The cervical vertebrae of Thalassemys hugii are much better preserved, even though they suffered strong lateral pressure. Taking postmortem deformation into account, these cervical vertebrae are also consistent with what is known in Plesiochelys bigleri: moderately long amphicoelous centrum, robust but low ventral keel, moderately high neural arch (notably posteriorly), and widely separated zygapophyses.

Thoracic vertebrae can best be observed in the articulated specimens MJSN BSY007-147 and MJSN TCH007-519, as well as in the disarticulated specimen MJSN SCR011-30. In these specimens, all sufficiently preserved thoracic vertebrae belong to the posterior part of the thoracic series (sixth thoracic vertebra and behind), and are biconcave, smoothly rounded ventrally and without keel: e.g., sixth thoracic vertebra in MJSN BSY007-147, seventh thoracic vertebra in MJSN TCH007-519, and the disarticulated thoracic vertebrae (probably 9th and 10th) of MJSN SCR011-30.

According to Bräm (1965), the thoracic vertebrae of Plesiochelys etalloni and C. jaccardi are keeled ventrally, but the best preserved available vertebrae all belong to the anterior part of the thoracic series (from first to fifth thoracic vertebrae): e.g., first thoracic vertebra of NMS 8723, fourth thoracic vertebra of NMS 8731, and fifth thoracic vertebra of MJSN TCH006-574. In the absence of complete thoracic series in either Plesiochelys etalloni or Plesiochelys bigleri, it is uncertain whether the condition in these two species differs or not.

One sacral and two caudal vertebrae of MJSN SCR011-30 are preserved. The sacral vertebra is short, and has two narrow prezygapophyses and a small keel on the centrum. The centrum of one caudal vertebra bears a robust ventral keel, similar to the cervical vertebrae. Poor preservation prevents any conclusion on the type of central articulation in the caudal vertebrae.

A still articulated series of caudal vertebrae is preserved in specimen NMS 8584 referred to Plesiochelys etalloni. As far as observable, their shape is consistent with what is known in Plesiochelys bigleri. Bräm (1965) described their centra as procoelous. However, the state of preservation of these centra does not allow to confirm Bräm’s observation.

Plesiochelys Etalloni

Fifteen specimens from Porrentruy are referred to Plesiochelys etalloni (Table 1). They are all represented by elements of the carapace and most of them also by the plastron. The shells MJSN BSY007-205, MJSN TCH005-332, and MJSN TCH006-574 are still articulated, the latter being by far the best preserved specimen (Fig. 16). Non-shell post-cranial material is only poorly preserved. For example, small remains of the scapula and pelvis are associated with MJSN BSY003-347, but the poor preservation impedes any anatomical comparisons.

Figure 16 MJSN TCH006-574, Plesiochelys etalloni (Kimmeridgian, Porrentruy, Switzerland).

Carapace in dorsal (A, B) and right lateral view (C); plastron in ventral view (D, E). Line width indicates natural borders (thick lines), bone sutures (medium lines), and fractures (thin lines); double lines indicate scale sulci. epi, epiplastron; pl, pleural scale; py, pygal.

As mentioned above, Plesiochelys bigleri and Plesiochelys etalloni mostly differ in their cranial anatomy. However, some characteristics of the shell allow to tell the two species apart. The main characters we used for this study are the thickness of neural and costal bones, and the presence and development of the epiplastral bulbs. In Plesiochelys bigleri, the epiplastral bulbs are reduced or absent (Fig. 7C), whereas they are usually well developed in Plesiochelys etalloni (Figs. 16D and 16E). The neural and costal bones of Plesiochelys etalloni are usually remarkable for their great relative thickness (Fig. 9D). Specimens of adult size regularly have neurals reaching 15–20 mm in thickness. Similarly sized specimens of Plesiochelys bigleri usually have a neural thickness ranging between 11 and 14 mm. However, there is a great deal of variation in both species, and the difference between the two is less obvious in juvenile specimens. Neural (and costal) thickness alone is therefore not always sufficient to discriminante between the two species. In order to test whether differences in neural thickness are significant in the two species, we measured 43 specimens from Porrentruy referred to Plesiochelys bigleri, Plesiochelys etalloni, or Plesiochelys sp. (see below).

The newly discovered shells of Plesiochelys etalloni show about the same range of variation as previously described for this species (Anquetin, Püntener & Billon-Bruyat, 2014). MJSN TCH006-574 is however remarkable in its extremely reduced fourth pleurals, which are restricted to the peripheral bones due to the great posterolateral development of the fourth vertebral scute (Fig. 16B). The fourth pleurals are similarly reduced in some specimens from Solothurn, but they always occupy at least a small part of the costals (e.g., NMS 8514 and NMS 8517; Anquetin, Püntener & Billon-Bruyat, 2014: figs. 2, 8). In MJSN TCH006-574, the 12th pair of marginal scutes is restricted to the pygal, which is a common variation in Plesiochelys bigleri (see above), but is unknown in other specimens referred to Plesiochelys etalloni (Anquetin, Püntener & Billon-Bruyat, 2014).

Neural Thickness in Plesiochelys

The neural length and thickness was measured on selected specimens (see Materials and Methods; Fig. 17B; Table 3). The scatter plot of mean length and thickness measurements reveals a relatively clear separation between Plesiochelys bigleri and Plesiochelys etalloni (Fig. 17A). This separation is mostly due to the proportionally increased neural thickness observed in Plesiochelys etalloni, as confirmed by the Mann–Whitney tests for mean thickness and length/thickness ratio (p < 0.0001, respectively). Mean neural length however is not significantly different in the two species (p = 0.1156; Fig. 17D).

Figure 17 Neural bone thickness in Plesiochelys etalloni and Plesiochelys bigleri.

Statistical analysis of mean neural length, mean neural thickness, and corresponding length/thickness ratio (see text) for 25 specimens of Plesiochelys bigleri (green), eight specimens of Plesiochelys etalloni (blue), and 10 indeterminate specimens (Plesiochelys sp.; gray) (all specimens from the Kimmeridgian of Porrentruy, Switzerland). (A) Mean length/thickness scatter-plot (specimen numbers are indicated for indeterminate specimens). (B) Length and thickness measurements on the fifth neural bone of specimen BSY006-307 (scale bar = 20 mm). (C) Discriminant histogram. (D–F) Box-and-whisker plots for mean length, mean thickness, and length/thickness ratio, respectively.

Table 3 Measurements used for the analysis of neural thickness in Plesiochelys spp.

Specimen	Identification	Mean length	Mean thickness	Ratio	
BSY006-307	Plesiochelys bigleri	60.62	12.75	4.75	
TCH007-252	Plesiochelys bigleri	56.99	10.15	5.62	
SCR011-140	Plesiochelys bigleri	57.48	10.76	5.34	
BSY009-815	Plesiochelys bigleri	57.12	13.77	4.15	
BSY007-257	Plesiochelys bigleri	45.27	9.61	4.71	
SCR011-148	Plesiochelys bigleri	52.89	12.53	4.22	
SCR011-413	Plesiochelys bigleri	54.98	11.99	4.59	
TCH006-1420	Plesiochelys bigleri	56.09	11.65	4.81	
SCR011-276	Plesiochelys bigleri	60.57	13.22	4.58	
VTT006-299	Plesiochelys bigleri	61.39	12.17	5.04	
TCH005-16	Plesiochelys bigleri	61.07	11.50	5.31	
TCH005-464	Plesiochelys bigleri	57.14	11.70	4.89	
TCH005-21	Plesiochelys bigleri	63.02	13.16	4.79	
SCR011-37	Plesiochelys bigleri	52.90	11.46	4.61	
BSY008-206	Plesiochelys bigleri	59.31	12.88	4.60	
SCR010-1279	Plesiochelys bigleri	65.20	11.59	5.62	
VTT006-579	Plesiochelys bigleri	54.98	11.87	4.63	
TCH006-145	Plesiochelys bigleri	52.32	11.34	4.61	
BSY009-892	Plesiochelys bigleri	60.16	12.85	4.68	
TCH005-819	Plesiochelys bigleri	60.40	12.16	4.97	
BSY006-326	Plesiochelys bigleri	54.37	10.79	5.04	
SCR010-1009	Plesiochelys bigleri	56.53	11.34	4.99	
SCR011-160	Plesiochelys bigleri	58.74	12.34	4.76	
SCR010-1196	Plesiochelys bigleri	55.12	12.52	4.40	
TCH006-767	Plesiochelys bigleri	50.46	12.62	4.00	
BSY009-694	Plesiochelys etalloni	57.77	14.83	3.90	
SCR011-415	Plesiochelys etalloni	48.07	13.61	3.53	
BSY006-347	Plesiochelys etalloni	56.45	14.70	3.84	
BSY007-205	Plesiochelys etalloni	60.21	15.41	3.91	
SCR008-33	Plesiochelys etalloni	55.01	17.76	3.10	
SCR010-382	Plesiochelys etalloni	49.05	12.82	3.83	
TCH007-265	Plesiochelys etalloni	52.57	14.89	3.53	
TCH007-505	Plesiochelys etalloni	54.88	14.64	3.75	
BSY009-310	Plesiochelys sp.	53.24	13.12	4.06	
TCH007-272	Plesiochelys sp.	48.59	11.64	4.18	
TCH005-817	Plesiochelys sp.	49.50	14.81	3.34	
SCR010-450	Plesiochelys sp.	51.98	11.80	4.40	
SCR010-479	Plesiochelys sp.	44.50	11.34	3.92	
SCR011-525	Plesiochelys sp.	47.86	12.81	3.74	
BSY009-619	Plesiochelys sp.	53.38	15.22	3.51	
BSY008-240	Plesiochelys sp.	50.52	10.22	4.95	
SCR011-111	Plesiochelys sp.	48.35	13.70	3.53	
BSY008-484	Plesiochelys sp.	57.80	14.77	3.91	
Notes:

Mean neural length, mean neural thickness, and mean length/mean thickness ratio measured for selected specimens referred to Plesiochelys bigleri, Plesiochelys etalloni, and Plesiochelys sp. Measurements are expressed in millimeters. See Table S1 for original measurements. All specimens are housed at the MJSN.

The discriminant analysis resulted in a relatively good classification of specimens (93.94% of individuals correctly identified; Fig. 17C). Only two specimens (MJSN BSY009-815 and MJSN TCH006-767) are incorrectly classified as Plesiochelys etalloni. These two specimens happen to be the ones that lie the closest to the Plesiochelys etalloni morphospace (Fig. 17A). The discriminant analysis also provided tentative classifications for the 10 indeterminate specimens (Plesiochelys sp.), which correspond fairly well with the conclusions we can draw based on the length/thickness scatter plot (Fig. 17A; Table 4). We consider the classification of only two of these indeterminate specimens to be dubious. MJSN BSY 009-310 is classified by the discriminant analysis as Plesiochelys etalloni, but this specimen actually lies on the margin of the Plesiochelys bigleri morphospace (Fig. 17A). SCR010-479 is classified by the discriminant analysis as Plesiochelys bigleri, but lies so far from the known morphospaces of Plesiochelys bigleri and Plesiochelys etalloni that this classification must be regarded with caution for the moment.

Table 4 Classification of indeterminate specimens.

Specimen	Scatter plot	Discriminant analysis	
BSY009-310	Plesiochelys bigleri	Plesiochelys etalloni	
TCH007-272	Plesiochelys bigleri	Plesiochelys bigleri	
TCH005-817	Plesiochelys etalloni	Plesiochelys etalloni	
SCR010-450	Plesiochelys bigleri	Plesiochelys bigleri	
SCR010-479	?	Plesiochelys bigleri	
SCR011-525	Plesiochelys etalloni	Plesiochelys etalloni	
BSY009-619	Plesiochelys etalloni	Plesiochelys etalloni	
BSY008-240	Plesiochelys bigleri	Plesiochelys bigleri	
SCR011-111	Plesiochelys etalloni	Plesiochelys etalloni	
BSY008-484	Plesiochelys etalloni	Plesiochelys etalloni	
Notes:

Comparison of the tentative classifications of indeterminate specimens (Plesiochelys sp.) based on the length/thickness scatter plot (Fig. 17A) and the discriminant analysis (Fig. 17C). All specimens are housed at the MJSN.

Neural thickness is therefore a good character to help differentiating between Plesiochelys bigleri and Plesiochelys etalloni. In the analyzed sample, mean neural thickness ranges from 9.61 to 13.77 mm (mean = 11.95 mm) in Plesiochelys bigleri, and from 12.82 to 17.76 mm (mean = 14.83 mm) in Plesiochelys etalloni (Fig. 17E). This difference between the two species is even more clearly expressed in the length/thickness ratio, which ranges from 4 to 5.65 (mean = 4.79) in Plesiochelys bigleri, and from 3.1 to 3.91 (mean = 3.67) in Plesiochelys etalloni (Fig. 17F).

Discussion

Alpha taxonomy

There is a dual issue with the identification and distinction of Plesiochelys bigleri and Plesiochelys etalloni. First, the two species are so closely related that differences in their shell and appendicular anatomy are minimal (see above). Second, each of the two species is known by tenths of shells from a single locality and horizon. These extensive collections reveal a great intraspecific variability in the two species (Anquetin, Püntener & Billon-Bruyat, 2014; this study). Therefore, differentiating the two species can be challenging.

The holotype (MJSN TCH007-252) and paratype (MJSN TCH006-1451) specimens of Plesiochelys bigleri are of paramount importance to establish the distinction between the two species. The isolated cranium MJSN TCH006-1451 (Figs. 5 and 6) exhibits a number of characteristics that clearly set it apart from Plesiochelys etalloni: reduced processus trochlearis oticum, reduced posterior flooring of the cavum acustico–jugulare by the pterygoid, clear prootic–opisthotic contact on the floor of the fossa temporalis superior, pila prootica not ossified, processus paroccipitalis extending posterolaterally, anterior foramen nervi abducentis opening ventral and slightly anteromedial to the base of the processus clinoideus, surface below the dorsum sellae sloping more gently anteriorly, and foramina anterius canalis carotici cerebralis widely separated. The preservation of the cranium associated with the holotype specimen (Fig. 4) is not as good as that of the paratype, but it also exhibits most of the above differences with Plesiochelys etalloni, in addition to the following: more rounded foramen nervi trigemini, and absence of midline contact of the exoccipital in the floor of the foramen magnum. Based on our experience of Late Jurassic coastal marine turtles, these differences cannot be attributed to intraspecific variations, sexual dimorphism, or ontogeny. Most importantly, the cranium of the holotype is associated with a near-complete carapace and partial plastron (Fig. 7). The cranium was found literally within the associated shell during preparation alongside elements of the appendicular skeleton. There is therefore no doubt regarding the natural state of this association. Interestingly, the holotype shell is remarkably similar to that of Plesiochelys etalloni, except for the much thinner neural and costal bones and for the absence of epiplastral bulbs.

Among the 80 relatively complete shells studied herein, 41 can be confidently referred to the new species Plesiochelys bigleri based on the reduced neural and costal thickness, the absence or great reduction of epiplastral bulbs, and a generally more quadrangular anterior plastral lobe. Among the remaining 39 shells, 15 exhibit features that are consistent with an identification as Plesiochelys etalloni, notably the great thickness of neural and costal bones and/or the presence of well-developed epiplastral bulbs. The remaining 24 shells are provisionally identified as Plesiochelys sp. because they lack sufficient diagnostic features. Among the 56 specimens referred either to Plesiochelys bigleri or to Plesiochelys etalloni, 33 with well-preserved neural bones were selected for a statistical analysis of neural thickness. This analysis confirmed that the mean thickness and length/thickness ratio were statistically different in the two species, with a mean length/thickness ratio of 4.79 for Plesiochelys bigleri and 3.67 for Plesiochelys etalloni (see above). The length/thickness ratio of neural bones (notably from neurals 2 to 5) is therefore an important additional feature to consider in order to differentiate these two species.

Is Plesiochelys bigleri also present in Solothurn?

The Solothurn turtle assemblage is diversified and slightly younger than the one from Porrentruy. However, the two localities share a number of species in common: Plesiochelys etalloni, Tropidemys langii, Thalassemys hugii, and Thalassemys bruntrutana (Rütimeyer, 1873; Bräm, 1965; Püntener et al., 2014; Püntener, Anquetin & Billon-Bruyat, 2015). Given the similarity observed between Plesiochelys bigleri and Plesiochelys etalloni in Porrentruy, the presence of Plesiochelys bigleri among Solothurn specimens referred to Plesiochelys etalloni would certainly not come as a surprise.

Six skulls from Solothurn are referred to Plesiochelys etalloni (NMS 8738, NMS 8739, NMS 8740, NMS 9145, NMS 40870, and NMS 40871; Gaffney, 1975a, 1976; Anquetin, Püntener & Billon-Bruyat, 2015). They all appear to belong to that species, although there may be some doubts regarding NMS 9145 which is associated with unprepared postcranial material including relatively thin costal bones. The latter specimen should be re-evaluated. Remaining specimens referred to Plesiochelys etalloni are otherwise often preserved as articulated shells, which complicates observation of neural and costal bones thickness in some cases and prevents a statistical analysis of neural bone thickness. Most of these specimens exhibit traits that are compatible with Plesiochelys etalloni, notably: relatively thick neurals or costal bones (e.g., NMS 8461, NMS 8515, NMS 8517, and NMS 8732), well-developed epiplastral bulbs (e.g., NMS 8533, NMS 8693, NMS 9150, NMS 9153, and NMS 9173), or both features at the same time (e.g., NMS 8582). In some specimens, neither of the two main distinguishing features can be observed due to preservation (e.g., NMS 8727). The attribution of other specimens might be questioned because one of the distinguishing feature is poorly expressed, whereas the other is impossible to check. For example, NMS 8731 has only moderately expressed epiplastral bulbs, and the thickness of the neurals and costals is difficult to evaluate (the costals seem to be relatively thin).

For the moment, none of the Plesiochelys specimens from Solothurn can be confidently referred to Plesiochelys bigleri. However, this material should definitely be re-evaluated in the future in light of the new material from Porrentruy.

Conclusion

Plesiochelys bigleri is a new plesiochelyid turtle known based on 41 relatively complete, but mostly disarticulated shells and two crania (one associated with a shell, and another one isolated). All of this material originates from a series of close by localities west of the small town of Courtedoux, near Porrentruy, Canton of Jura, Switzerland. Most of the specimens were collected from a single stratigraphically limited horizon, the Lower Virgula Marls, dated from the early late Kimmeridgian. A few additional specimens were found in two underlying horizons, the Corbis Limestones and Banné Marls, dated from the late early Kimmeridgian.

The shell morphology of Plesiochelys bigleri is remarkably similar to that of Plesiochelys etalloni, a species known based on tenths of shells and several crania from the Kimmeridgian of Switzerland, France, Germany, and England. These two closely related species however differ in the thickness of the neural and costal bones of the carapace (a difference that is statistically tested herein), and the presence and development of the epiplastral bulbs in the plastron. Differentiating the two species based only on shell morphology can be challenging in some incomplete or juvenile individuals. The two species co-occur in Porrentruy and 24 shells (30% of the shells referable to Plesiochelys) cannot be identified at the species level as a result of this great similarity. However, the two species are more easily separated based on cranial morphology. Actually, Plesiochelys bigleri exhibits cranial features that clearly set it apart from Plesiochelys etalloni and other plesiochelyids, such as: a rounded foramen nervi trigemini, a shallow pterygoid fossa, a reduced processus trochlearis oticum, the absence of ossification of the pila prootica, the surface below the dorsum sellae sloping rather gently anteroventrally, and the widely separated foramina anterius canalis carotici cerebralis.

For the moment, Plesiochelys bigleri is known only in Porrentruy. However, Solothurn and Porrentruy share several species of turtles in common (Plesiochelys etalloni, Tropidemys langii, Thalassemys hugii, and Thalassemys bruntrutana), and these species are also known in the Kimmeridgian of southern England (Püntener et al., 2014; Püntener, Anquetin & Billon-Bruyat, 2015; Anquetin & Chapman, 2016). Finding Plesiochelys bigleri in other localities would therefore not come as a surprise.

The abundant material from Solothurn and Porrentruy referred to Plesiochelys etalloni and Plesiochelys bigleri illustrates the extent of intraspecific variability in these two species. Although these results may not be blindly transposable to other groups of turtles, they represent an important point of comparison for other studies on Mesozoic turtle diversity.

Supplemental Information

Supplemental Information 1 MJSN TCH007-252, holotype of Plesiochelys bigleri (Kimmeridgian, Porrentruy, Switzerland).

3D model of the cranium.

Click here for additional data file.

Supplemental Information 2 MJSN TCH006-1451, paratype of Plesiochelys bigleri (Kimmeridgian, Porrentruy, Switzerland).

3D model of the cranium.

Click here for additional data file.

Supplemental Information 3 Original measurements of neural length and thickness.

Thickness was measured on the left and right side, respectively, and approximately at the middle of each neural. Measurements are expressed in millimeters. All specimens are housed at the MJSN.

Click here for additional data file.

We thank Loïc Bocat (excavation), Pierre Bigler, Renaud Roch and Sébastien Bergot (preparation team), Bernard Migy and Olivier Noaillon (photographs), Pierre Widder (scientific drawings), Apolline Lefort (discussion on stratigraphy) and the whole Paleontology A16 team. We furthermore thank Silvan Thüring (NMS) and Loïc Costeur (NMB) for access to collections in their care. Editor Mark Young and the reviewers Adá Pérez-García and Márton Rabi provided very helpful comments to improve the manuscript.

Institutional Abbreviations

MAJ Musée d’archéologie du Jura, Lons-le-Saunier, France

MJSN JURASSICA Museum, Porrentruy, Switzerland

NHMUK Natural History Museum, London, UK

NMB Naturhistorisches Museum Basel, Switzerland

NMS Naturmuseum Solothurn, Switzerland

OUMNH Oxford University Museum of Natural History, Oxford, UK

PIMUZ Paläontologisches Institut und Museum, Universität Zürich, Switzerland.

Locality Abbreviations

BSY Bois de Sylleux, Courtedoux, near Porrentruy, Switzerland

CRT Crat, Chevenez, near Porrentruy, Switzerland

SCR Sur Combe Ronde, Courtedoux, near Porrentruy, Switzerland

TCH Tchâfoué, Courtedoux, near Porrentruy, Switzerland

VTT Vâ Tche Tchâ, Courtedoux, near Porrentruy, Switzerland.

Additional Information and Declarations

Competing Interests

Author Contributions

Data Availability

New Species Registration

Jérémy Anquetin is an Academic Editor for PeerJ.

Christian Püntener conceived and designed the experiments, performed the experiments, analyzed the data, wrote the paper, prepared figures and/or tables, and reviewed drafts of the paper.

Jérémy Anquetin conceived and designed the experiments, performed the experiments, analyzed the data, wrote the paper, prepared figures and/or tables, and reviewed drafts of the paper.

Jean-Paul Billon-Bruyat conceived and designed the experiments, analyzed the data and reviewed drafts of the paper.

The following information was supplied regarding data availability:

Anquetin, Jérémy; Migy, Bernard; Noaillon, Olivier (2017): 3D model of cranium MJSN TCH007-252 (Plesiochelys bigleri). figshare. https://dx.doi.org/10.6084/m9.figshare.4806922.v1.

Anquetin, Jérémy; Migy, Bernard; Noaillon, Olivier (2017): 3D model of cranium MJSN TCH006-1451 (Plesiochelys bigleri). figshare. https://dx.doi.org/10.6084/m9.figshare.4785484.v1.

Anquetin, Jérémy (2017): 3D model of pelvis MJSN BSY006-307 (Plesiochelys bigleri). figshare. https://dx.doi.org/10.6084/m9.figshare.4806961.v1.

The following information was supplied regarding the registration of a newly described species:

Publication LSID:

urn:lsid:zoobank.org:pub:C5AE9DE8-9911-4CFD-AD09-65850C35BDEC

Plesiochelys bigleri LSID:

urn:lsid:zoobank.org:act:9A8EF46E-7DAA-4F5B-B727-58C559BA503C.

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
