# Peer review of "The comparative osteology of Plesiochelys bigleri n. sp., a new coastal marine turtle from the Late Jurassic of Porrentruy (Switzerland)"

_PeerJ, doi:10.7717/peerj.3482_

## Round 0.1 · original submission · Minor Revisions

Dear authors,

I apologise for the latest of the review decision. Based on the comments of both reviewers I have accepted their decision of 'minor revisions'.

Please take note of the comments from reviewer two, and the suggestion your new taxon may represent intraspecific variation.

·

Basic reporting

no comment

Experimental design

no comment

Validity of the findings

no comment

Additional comments

I consider this manuscript as very interesting and very well written. I recommend its publication in this Journal. I have indicated some minor comments and suggestions on the pdf.

I'm at the disposal of the authors and the editors for any question.

Sincerely,

Adán Pérez-García

·

Basic reporting

no comment

Experimental design

no comment

Validity of the findings

This is a great description of important Mesozoic fossil turtle material. The authors propose to erect a new species of Plesiochelys besides P. etalloni from the same locality and horizon. However, I don't find the data convincing enough for identifying a separate species from Plesiochelys etalloni (see below). I suggest to publish this description with out erecting a new species and assigning the material to P. etalloni (or Plesiochelys sp. if this is the authors' preference).

Additional comments

After carefully reading the manuscript twice, I must conclude that I am puzzled why the authors propose to erect a new species.

Most characters used for distinguishing the proposed new species from etalloni are cranial. Only two fragmentary skulls are referred to this proposed new taxon and both of them are poorly preserved: crushed, deformed, damaged and incomplete. The authors list cranial characters that are said to be separate the proposed new taxon from Plesiochelys etalloni but it is important to note that none of them are discrete characters. Instead, these characters pertain to minor differences in the relative size, orientation, shape, ossification, length or depth of certain basicranial structures. Usually, such minor differences can be explained by preservation (the skulls described are deformed and damaged) or intraspecific (incl. ontogenetic) variation.

I don't really understand how the authors can rely on these minor differences when there are only two! (badly preserved and only partly-overlapping) fragmentary skulls are available for this purported species.

The authors report on other differences in the description but these are either variable or not known in etalloni or may be preservational differences.

The postcranial characters used to distinguish the proposed new taxon from etalloni are likewise problematic. The authors themselves point out in their description that the same conditions may be present in some individuals of etalloni due to the extensive variation within that species.

Neural thickness: a statistical analysis was performed on neural bone thickness in order to test for morphological clusters within the population collected at the locality. Unfortunately, I am not able to see two distinct clusters in their figure: instead minor overlap is present with etalloni and "bigleri" and great overlap between these two and Plesiochelys sp. Why Plesiochelys sp. specimens are ID-d as such and not some as etalloni and others as "bigleri"? I find this unjustified and sort of cherry-picking. Furthermore, the low sample size (n=43) makes me expect that the addition of further specimens to the analysis could very well result in filling the space between etalloni and "bigleri".

Note that even if there were two clusters there is no way to objectively interpret whether this is because of sexual dimorphism or two taxa.

To test the utility of neural thickness, the authors would need to analyze the etalloni population from Solothurn alone and in combination with the material from A16. But even in this case the sample size may be too low.

Another more rigorous test of taxonomic status would be a specimen level phylogeny of Plesiochelys.

I think this paper would be a more important contribution with out naming a new species. Good anatomical descriptions are always needed and this is certainly one of them (illustration of appendicular elements are particularly welcome). I see no justification for naming a new species and I think it would only cause taxonomic problems in the future. The differences between the two species are such nuances (the authors themselves admit it) that it will be very difficult to distinguish them at other localities (again the authors themselves admit this). P. etalloni is widely used in phylogenetic matrices but with the new species some of the material used for scoring would have to be re-evaluated and may turn out to be unclear what specimens can be referred to etalloni with confidence. It also could lead to a lower number of "scoreable" characters for etalloni. The new species makes any analysis on the evolution of etalloni unnecessarily more complicated. I wonder if the authors are aware of this.

P. etalloni is a phylogenetically important taxon, the best known plesiochelyid with a relatively good understanding of intraspecific variation - besides Gaffney, largely thanks to these very authors' earlier efforts. I am asking the authors: what do we benefit from changing this situation (= making "our life" more complex) especially when the change is possible to avoid?

A finale note: I assume that the authors are aware that two sympatric closely related species infer a rather complex evolutionary scenario. There is a general consensus on that the most common way of speciation in animals is allopatric speciation. That being said, sympatric distribution of two closely related species infers an initial geographic isolation (speciation) and subsequent merge of the distributional range of the two (recently split) species. Is this really what the authors intend to propose? Please note that you are indirectly doing so. There are reports on such cases among living species but those are based on solid phylogenetic evidence. In any case, this is always a more complex scenario than two sympatric species that are not closely related (i.e. if "bigleri" is indeed valid it may not be the best choice to refer it to the same genus as etalloni).

In summary, I suggest not to erect a new species but publish the description with further discussion on intraspecific variability in etalloni. That would make a very valuable contribution. Perhaps a table summarizing the temporal and spatial distribution of Late Jurassic eucryptodires from Central-Europe would be also welcome.

---

## Round 0.2 · Minor Revisions

Dear authors,

After sending the revised manuscript to the major reviewer from the previous round, I have accepted the decision of 'minor review'.

Should you decide to keep the new taxon, can you look into the comments given by the reviewer and examine the characters used and your diagnosis.

·

Basic reporting

no comment

Experimental design

no comment

Validity of the findings

I remain sceptical with the validity of the proposed taxon.

Additional comments

I thank the authors for addressing some of my concerns. I, however, remain sceptical with the validity of this taxon and below I provide my critics for each of the characters that the authors consider diagnostic. Based on my review of photos, papers, and personal observation of all available comparative material of Plesiochelys spp., the vast majority of these characters cannot be objectively considered diagnostic (mostly variable within etalloni or present in planiceps as well) and the remaining few are hard to reproduce. Whenever I could I provided reference or specimen number for my claims. I am attaching these comments as a separate word file, perhaps easier to follow the discussion that way. I hope this paper is going to be published (it is a great description) although I think the taxonomic assignment should be left open.

“Diagnosis. Plesiochelys bigleri differs from other Plesiochelys spp. in a lower temporal skull roof,

What does this mean ? Please rephrase. If you mean shallower emargination then are you sure the specimen is intact? It does not seem so. If you mean the height of the skull then make sure you account for crushing and deformation.

...a shallower pterygoid fossa,

Note that the deepness of this fossa is variable among extant turtles. Also it looks deeper in the specimen in Fig. 4 than in Fig 5.

...a reduced processus trochlearis oticum,

I disagree with this claim. Cf. your Fig. 4 with Anquetin 2015 Fig. 4 of etalloni. There is no difference. In the specimen in Fig. 5. the processus is apparently damaged and incomplete.

...a more rounded foramen nervi trigemini,

Plesiochelys planiceps has the same character: Gaffney 1979 Fig. 9. Also etalloni: SM 135 (NMS 8739).

...an anterior foramen nervi abducentis opening anteromedially to the base of the processus clinoideus,

Same as in etalloni. Cf. your Fig. 4 with Anquetin 2015 Fig. 4-

...and foramina anterius canalis carotici cerebralis opening more anteriorly relative to the level of the dorsum sellae.

This may be a minor difference but is hard to reproduce. Are you sure that the dorsum sella in your Fig. 5 is not damaged? Do you think there are enough specimens of etalloni preserving this region for assessing intraspecific variability? In NHMUK R3370 it is apparently more posterior than in SM 134 or NMS 40871 of etalloni. So there is variability and perhaps there are specimens where it is more anterior? In your two new specimens are the foramina in the same position?

...In addition, Plesiochelys bigleri differs from Plesiochelys planiceps in a smaller size,

Perhaps due to different ontogenetic stage or sexual dimorphism?

...a lower lingual ridge on the maxilla,

Please note that this is variable within a single species of extant turtles and there are only one specimen of “bigleri” and very few of planiceps (only 1?) showing this so it is difficult to assess variability. Also the "bigleri" specimen has an incompletely preserved lingual ridge which makes comparison with planiceps difficult if not impossible (based on the photos, I don't even see the difference).

...a parietal-quadrate contact posterior to the foramen nervi trigemini,

In your figure 4 this contact is illustrated as being ambigous on the right side (not exposed on the left in lateral view). Note that this character is not illustrated for planiceps nor properly described in the literature. This contact is always difficult to observe in fossils and I wonder how many specimens of etalloni preserved this region well enough so you can rule out intraspecific variation. (For planiceps, as you know, there is only one specimen available).

...a less developed processus trochlearis oticum,

See above

...a superficial canalis caroticus internus that may have remained partly open ventrally,

This is a question of preservation in other specimens and certainly difficult to interpret in yours where apparently damaged. Isn't this always close to the surface in Plesiochelys and related taxa? The fact that a lot of specimen exposes this canal clearly indicates that it is close to the surface or exposed naturally in most species.

...and an absent or reduced contribution of the exoccipital to the condylus occipitalis

Absent or reduced ? Also an absent condition is very uncommon in turtles. Corsochelys is reported to have it absent but infact the specimen does not show this. Are you sure it is not fused (quite common)? Again, only one specimen of planiceps is known that is not described in detail.

..., and from Plesiochelys etalloni in a less extensive flooring of the cavum acustico-jugulare by the pterygoid,

I honestly cannot reproduce this difference when compared to the range of morphologies present in the different specimens of etalloni. E.g., NHMUK R3370 or MH 435 or SM 136 have this region largely exposed as well.

...the absence of complete ossification of the pila prootica,

In your Fig. 5 this region looks damaged. Is it possible that it is broken in both of your specimens?

...a processus paroccipitalis extending posterolaterally,
This was difficult to reproduce what you mean but now I understand. However, this is also present in SM 135 & 136 (NMS 8739 & 8740) and I would say also in SM 134 (8738) if it was complete of etalloni.

...a reduced neural and costal bone thickness,

I quote you : “We do not say that these results confirm that we have two species.” In this case, I suggest removing it from the diagnosis.

...absent or poorly developed epiplastral bulbs,

Also absent or poorly developed in the holotype and MH 435 of etalloni.

...and a more quadrangular anterior margin of the anterior plastral lobe.”

I quote you : "The anterior plastral lobe of Pl. bigleri is often somewhat quadrangular in outline (e.g.,784 MJSN SCR010-1196, MJSN SCR011-140; Figs. 8K–8L), but can also be rounded (e.g., MJSN 785 CRT007-2), or even pointed (e.g., MJSN TCH006-1420; Figs. 8C–8D) in some specimens." And: " In P. etalloni the anterior plastral lobe usually has a rather rounded, sometimes pointed, anterior 787 outline (Anquetin, Püntener & Billon-Bruyat, 2014)."

This sounds to me that there is a continum from quadrangular to rounded to pointed. In addition there are not many specimens of etalloni preserving this region completely. Rounded may well be the most common morphotype in etalloni but compare your Fig. 7 with Anquetin et al. (2014) Fig. 8 K and W. It is not possible to distinguish them based on this character.


Other comments:
MY COMMENT FROM 1ST ROUND: Neural thickness: a statistical analysis was performed on neural bone thickness in order to test for morphological clusters within the population collected at the locality. Unfortunately, I am not able to see two distinct clusters in their figure: instead minor overlap is present with etalloni and "bigleri" and great overlap between these two and Plesiochelys sp. Why Plesiochelys sp. specimens are ID-d as such and not some as etalloni and others as "bigleri"? I find this unjustified and sort of cherry-picking. Furthermore, the low sample size (n=43) makes me expect that the addition of further specimens to the analysis could very well result in filling the space between etalloni and "bigleri".
Note that even if there were two clusters there is no way to objectively interpret whether this is because of sexual dimorphism or two taxa.
To test the utility of neural thickness, the authors would need to analyze the etalloni population from Solothurn alone and in combination with the material from A16. But even in this case the sample size may be too low.

AUTHOR’S RESPONSE: This small statistical analysis was not conducted as a way to prove we were justified in creating two species. Instead, we designed it as a way to test whether the differences in neural thickness we though we perceived when observing the material were statistically supported. We feed the analysis with our identification based on morphological observations. Specimens classified as Plesiochelys sp. were identified as such because no other characters were available to refer them to any of the two species at hand. They usually consist of relatively incomplete specimens. The analysis tells us that the two groups we identified have statistically different means for thickness and length/thickness ratio. We do not say that these results confirm that we have two species. However, we point that specimens we identify as Pl. bigleri based on other cranial and/or postcranial characters are also characterized by a lower neural thickness. For reasons explained above, we consider that these differences are best explained by the two-species hypothesis.
We agree that the sample size is relatively small, but not uncommonly small for fossil vertebrates from a single locality. As any other statistical analysis, results are prone to change with the inclusion of further specimens, but this is no argument to disregard the result of the present analysis. Finally, we would have analyzed the Solothurn material if it was possible, but the preservation of the material prevents a similar analysis.

MY RESPONSE: I repeat some of my earlier concerns since the authors did not explicitly respond to them: Note that even if there were two clusters there is no way to objectively interpret whether this is because of sexual dimorphism or two taxa since you only have a single skull-shell association. Unfortunately, I am not able to see two distinct clusters in their figure: instead minor overlap is present with etalloni and "bigleri" and great overlap between these two and Plesiochelys sp. Why Plesiochelys sp. specimens are ID-d as such and not some as etalloni and others as "bigleri"? I find this unjustified and sort of cherry-picking.



MY COMMENT FROM 1ST ROUND: I think this paper would be a more important contribution with out naming a new species. Good anatomical descriptions are always needed and this is certainly one of them (illustration of appendicular elements are particularly welcome). I see no justification for naming a new species and I think it would only cause taxonomic problems in the future. The differences between the two species are such nuances (the authors themselves admit it) that it will be very difficult to distinguish them at other localities (again the authors themselves admit this). P. etalloni is widely used in phylogenetic matrices but with the new species some of the material used for scoring would have to be re-evaluated and may turn out to be unclear what specimens can be referred to etalloni with confidence. It also could lead to a lower number of "scoreable" characters for etalloni. The new species makes any analysis on the evolution of etalloni unnecessarily more complicated. I wonder if the authors are aware of this.
P. etalloni is a phylogenetically important taxon, the best known plesiochelyid with a relatively good understanding of intraspecific variation - besides Gaffney, largely thanks to these very authors' earlier efforts. I am asking the authors: what do we benefit from changing this situation (= making "our life" more complex) especially when the change is possible to avoid?

AUTHORS RESPONSE: We are plenly aware that the situation is complex. However, the point is not to wonder whether this new material makes the situation more complicated or not.

MY RESPONSE: I think to certain degree it is because there is very little evidence (if any) for erecting a new species and therefore I would rather wait until more material comes to surface. That will decide the issue.

MY COMMENT FROM 1ST ROUND: A finale note: I assume that the authors are aware that two sympatric closely related species infer a rather complex evolutionary scenario. There is a general consensus on that the most common way of speciation in animals is allopatric speciation. That being said, sympatric distribution of two closely related species infers an initial geographic isolation (speciation) and subsequent merge of the distributional range of the two (recently split) species. Is this really what the authors intend to propose? Please note that you are indirectly doing so. There are reports on such cases among living species but those are based on solid phylogenetic evidence. In any case, this is always a more complex scenario than two sympatric species that are not closely related (i.e. if "bigleri" is indeed valid it may not be the best choice to refer it to the same genus as etalloni).

AUTHORS RESPONSE: Geographic isolation is not the only driver for speciation. Research on these Late Jurassic coastal marine turtles shows that this is a highly diversified group that rapidly radiated into shallow carbonate platform environments during the Late Jurassic. Adaptation to different ecological niches when a group invades a new environment is also a powerful driver for speciation and may explain the situation at hand.

MY RESPONSE: It is not the only but certainly the most common. If you are hypothesising adaptive radiation as driving the speciation in the case of “bigleri” then this should be backed by some evidence. Also, this is not entirely clear for me on what basis this taxon or specimens are referred to the genus Plesiochelys. This should be explicitly explained in the discussion (ideally a phylogenetic analysis is provided for demonstrating the generic affinity).

---

## Round 0.3 · accepted · Accept

Dear authors,

Many thanks for your revised manuscript. After reading it, I have accepted it for publication in PeerJ.

Once again, thank you for submitting your manuscript to PeerJ and I hope you will use us again as your publication venue.

If we need to clarify any details required to move the manuscript forward, then our production staff will get in touch with you. Otherwise, a proof will be forthcoming shortly for your review.

Congratulations and thank you for your submission.